# Composing Partial Differential Equations with Physics-Aware Neural Networks

## Abstract

We introduce a compositional physics-aware neural network (FINN) for learning spatiotemporal advection-diffusion processes. FINN implements a new way of combining the learning abilities of artificial neural networks with physical and structural knowledge from numerical simulation by modeling the constituents of partial differential equations (PDEs) in a compositional manner. Results on both one- and two-dimensional PDEs (Burger's, diffusion-sorption, diffusion-reaction, Allen-Cahn) demonstrate FINN's superior modeling accuracy and excellent out-of-distribution generalization ability beyond initial and boundary conditions. With only one tenth of the number of parameters on average, FINN outperforms pure machine learning and other state-of-the-art physics-aware models in all cases—often even by multiple orders of magnitude. Moreover, FINN outperforms a calibrated physical model when approximating sparse real-world data in a diffusion-sorption scenario, confirming its generalization abilities and showing explanatory potential by revealing the unknown retardation factor of the observed process.

## 1 Introduction

Artificial neural networks (ANNs) are considered universal function approximators (Cybenko, 1989). Their effective learning ability, however, greatly depends on domain and task-specific pre-structuring and methodological modifications referred to as inductive biases (Battaglia et al., 2018). Typically, inductive biases limit the space of possible models by reducing the opportunities for computational shortcuts, which can lead to erroneous implications derived from a potentially limited dataset (overfitting). The recently evolving field of physics-informed machine learning employs physical knowledge as inductive bias providing huge generalization advantages in contrast to pure machine learning (ML) in physical domains (Raissi et al., 2019). While numerous approaches have been introduced to augment ANNs with physical knowledge, these methods either do not allow the incorporation of explicitly defined physical equations (Long et al., 2018; Seo et al., 2019; Guen & Thome, 2020; Li et al., 2020a; Sitzmann et al., 2020) or cannot generalize to other initial and boundary conditions than those encountered during training (Raissi et al., 2019).

In this work, we present the finite volume neural network (FINN) model—a physics-aware neural network structure adhering to the idea of spatial and temporal discretization in numerical simulation. FINN consists of multiple neural network modules that interact in a distributed, compositional manner (Battaglia et al., 2018; Lake et al., 2017; Lake, 2019). The modules are designed to account for specific parts of advection-diffusion equations, a class of partial differential equations (PDEs). This modularization allows to combine two advantages that are not yet met by state-of-the-art models: the explicit incorporation of physical knowledge and the generalization over initial and boundary conditions. To the best of our knowledge, FINN's ability to adjust to different initial and boundary conditions and to explicitly learn constitutive relationships and reaction terms is unique, yielding excellent out-of-distribution generalization. The core contributions of this work are:

- Introduction of FINN, a physics-aware neural network model, explicitly designed to generalize over initial and boundary conditions, demonstrating excellent generalization ability.

- Evaluation of state-of-the-art pure ML and physics-aware models, contrasted to FINN on one- and two-dimensional benchmarks, demonstrating benefits of explicit model design.

- Application of FINN to a real-world contamination-diffusion problem, verifying its applicability to real, spatially and temporally constrained training data.

## 2 RELATED WORK

**Non-physics-aware ANN architectures**  Pure ML models that are designed for spatiotemporal data processing can be separated into temporal convolution (TCN, Kalchbrenner et al., 2016) and recurrent neural networks. While the former perform convolutions over space and time, representatives of the latter, e.g., convolutional LSTM (ConvLSTM, Shi et al., 2015) or DISTANA (Karlbauer et al., 2019), aggregate spatial neighbor information to further process the temporal component with recurrent units. Since pure ML models do not adhere to physical principles, they require enormous amounts of training data and parameters in order to approximate a desired physical process; but still are not guaranteed to behave consistently outside the regime of the training data.

**Physics-aware ANN architectures**  When designed to satisfy physical equations, ANNs are reported to have greater robustness in terms of physical plausibility. For example, the physics-informed neural network (PINN, Raissi et al., 2019) consists of an MLP that satisfies an explicitly defined PDE with specific initial and boundary conditions, using automatic differentiation. However, the remarkable results beyond the time steps encountered during training are limited to the very particular PDE and its conditions. A trained PINN can actually not be applied to other different initial and boundary conditions, which limits its applicability in real-world scenarios.

Other methods by Long et al. (PDENet, 2018), Guen & Thome (PhyDNet, 2020), or Sitzmann et al. (SIREN, 2020) learn the first $n$ derivatives to achieve a physically plausible behavior, but lack the option to include physical equations. The same limitation applies when operators are learned to approximate PDEs (Li et al., 2020a;b) , or when physics-aware graph neural networks are applied (Seo et al., 2019). Yin et al. (APHYNITY, 2020) propose to approximate equations with an appropriate physical model and to augment the result by an ANN, preventing the ANN to approximate a distinct part within the physical model. For more comparison to related work, please refer to subsection A.1.

In summary, none of the above methods can explicitly learn particular constitutive relationships or reaction terms while simultaneously generalizing beyond different initial and boundary conditions.

## 3 FINITE VOLUME NEURAL NETWORK (FINN)

**Problem formulation**  Here, we focus on modeling spatiotemporal physical processes. Specifically, we consider systems governed by advection-diffusion type equations (Smolarkiewicz, 1983):

$$\frac{\partial u}{\partial t} = D(u)\frac{\partial^2 u}{\partial x^2} - v(u)\frac{\partial u}{\partial x} + q(u), \tag{1}$$

where $u$ is the quantity of interest, $t$ is time, $x$ is the spatial coordinate, $D$ is the diffusion coefficient, $v$ is the advection velocity, and $q$ is the source/sink term. Eq. 1 can be partitioned into three parts: the storage term, the flux terms, and the source/sink term. The storage term $\frac{\partial u}{\partial t}$ describes the change of the quantity $u$ over time. The flux terms are the advective flux $v(u)\frac{\partial u}{\partial x}$ and the diffusive flux $D(u)\frac{\partial^2 u}{\partial x^2}$. Both calculate the amount of $u$ exchanged between neighboring positions. The source/sink term $q(u)$ describes the generation or elimination of the quantity $u$. Eq. 1 is a general form of PDE with up to second order spatial derivative, but it has a wide range of applicability due to the flexibility of defining $D(u)$, $v(u)$, and $q(u)$ as different functions of $u$, as is shown by the numerical experiments in this work.

The finite volume method (FVM, Moukalled et al., 2016) discretizes a simulation domain into control volumes ($i = 1, \ldots, N_x$), where exchange fluxes are calculated using a surface integral (Riley et al., 2009). In order to match this structure in FINN, we introduce two different kernels, which are (spatially) casted across the discretized control volumes: the flux kernel, modeling the flux terms, and the state kernel, modeling the source/sink term as well as the storage term. The overall FINN architecture is shown in Figure 1.

**Flux kernel**  The flux kernel $\mathcal{F}$ approximates the surface integral for each control volume $i$ with boundary $\Omega$ by a composition of multiple subkernels $f_j$, each representing the flux through a discretized surface element $j$:

$$\mathcal{F}_i = \sum_{j=1}^{N_{s_i}} f_j \approx \oint_{\omega \subseteq \Omega} \left( D(u)\frac{\partial^2 u}{\partial x^2} - v(u)\frac{\partial u}{\partial x} \right) \cdot \hat{n}\, d\Gamma, \tag{2}$$

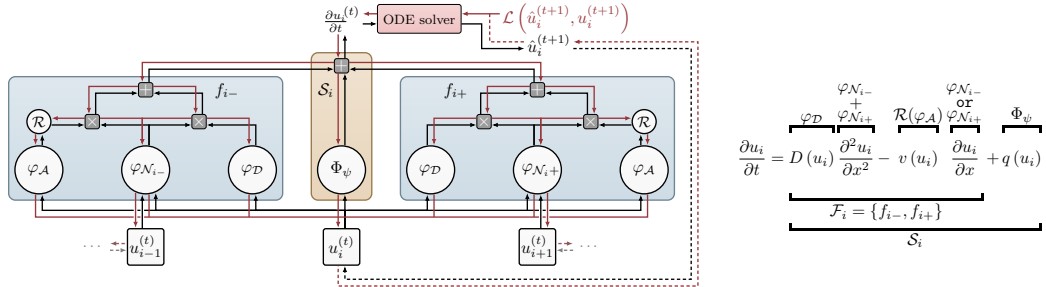

Figure 1: Flux and state kernels in FINN for the one-dimensional case (left); detailed assignment of the individual modules with their contribution to Eq. 1 (right). Read lines indicate gradient flow.

where $N_{s_i}$ is the number of discrete surface elements of control volume $i$, $\omega$ is a continuous surface element (a subset of $\Omega$), $f_j$ are subkernels (which are realized as feedforward network modules), and $\hat{n}$ is the unit normal vector pointing outwards of $\omega$. In our exemplary one-dimensional arrangement, two subkernels $f_{i-}$ and $f_{i+}$ (see Figure 1) contain the modules $\varphi_\mathcal{D}$, $\varphi_\mathcal{A}$, and $\varphi_\mathcal{N}$ ($\varphi_{\mathcal{N}_{i-}}$ or $\varphi_{\mathcal{N}_{i+}}$ for $f_{i-}$ and $f_{i+}$, respectively). The module $\varphi_\mathcal{N}$ is a linear layer with the purpose to learn the numerical FVM stencil. More specifically, the module $\varphi_{\mathcal{N}_{i-}}$ takes the inputs $u_i$ and its neighbor $u_{i-1}$, while $\varphi_{\mathcal{N}_{i+}}$ takes the inputs $u_i$ and $u_{i+1}$, and output the approximation of the spatial derivative $\frac{\partial u}{\partial x}$. This signifies that the weight of $\varphi_\mathcal{N}$ should amount to $[-1, 1]$ with respect to $[u_i, u_{i-1}]$ and $[u_i, u_{i+1}]$ (i.e. simple difference between neighboring control volumes) in ideal one-dimensional problems.

The other modules $\varphi_\mathcal{A}$ and $\varphi_\mathcal{D}$ are responsible for advective and diffusive fluxes, respectively. Both modules receive only $u_i$ as input. Moreover, the module $\mathcal{R}$ applies an upwind differencing scheme, which prevents numerical instability in the first order spatial derivative calculation (Versteeg & Malalasekera, 1995). It ensures that calculation of the advective fluxes using $\varphi_\mathcal{A}$ is performed only on one control volume surface (either left or right, and not both at the same time), whereas calculation of the diffusive fluxes using $\varphi_\mathcal{D}$ is performed on both surfaces. This means, for the advective flux calculation, the summation of the numerical stencil from both $f_{i-}$ and $f_{i+}$ will only lead to $[-1, 1]$, which corresponds to either $[u_i, u_{i-1}]$ when $\varphi_\mathcal{A} > 0$ or $[u_i, u_{i+1}]$ when $\varphi_\mathcal{A} < 0$ (i.e. only first order spatial derivative). For the diffusive flux calculation, the summation of the numerical stencil will lead to the classical one-dimensional numerical Laplacian with $[1, -2, 1]$ corresponding to $[u_{i-1}, u_i, u_{i+1}]$, that is, the second order spatial derivative, because the calculation of $\varphi_\mathcal{D}$ is performed on both surfaces (see subsection A.3 for more details). Accordingly, module $\mathcal{R}$ generates the inductive bias to make $\varphi_\mathcal{A}$ only approximate the advective, and $\varphi_\mathcal{D}$ the diffusive flux.

For the advective flux calculation, if the velocity $v$ is a function of $u$, the module $\varphi_\mathcal{A}$ approximates this dependence using a feedforward neural network, i.e. $\varphi_\mathcal{A}(u) \approx v(u)$. Otherwise, $\varphi_\mathcal{A}$ is a scalar value $\varphi_\mathcal{A} \equiv v$, which can be also set as a learnable parameter. The output of $\varphi_\mathcal{A}$ is activated by the following case-sensitive ReLU $\mathcal{R}$:

$$\mathcal{R}(\varphi_\mathcal{A}) = \begin{cases} \mathrm{ReLU}(\varphi_\mathcal{A}), & \text{on } f_{i-}, \\ -\mathrm{ReLU}(-\varphi_\mathcal{A}), & \text{on } f_{i+}, \end{cases} \tag{3}$$

and then multiplied with the output of $\varphi_\mathcal{N}$ (i.e. $\frac{\partial u}{\partial x}$) to obtain the corresponding advective flux at the surface. Similarly for the diffusive fluxes, if the diffusion coefficient $D$ depends on $u$, the module $\varphi_\mathcal{D}$ takes $u$ as an input to learn the function $\varphi_\mathcal{D}(u) \approx D(u)$ using a feedforward neural network. Otherwise, $\varphi_\mathcal{D}$ is a scalar value $\varphi_\mathcal{D} \equiv D$, which can also be set as a learnable parameter. The output of $\varphi_\mathcal{D}$ is not activated by the module $\mathcal{R}$, but is directly multiplied with the output of $\varphi_\mathcal{N}$ to obtain the corresponding diffusive flux at the surface. Consequently, integration of subkernels $f_j$ in Eq. 2 and substitution of Eq. 3 leads to the following representation:

$$\mathcal{F}_i = \begin{cases} \varphi_\mathcal{N}(u_i, u_{i-1})[\varphi_\mathcal{D} - \varphi_\mathcal{A}]_{f_{i-}} + \varphi_\mathcal{N}(u_i, u_{i+1})[\varphi_\mathcal{D}]_{f_{i+}}, & \text{if } \varphi_\mathcal{A} > 0, \\ \varphi_\mathcal{N}(u_i, u_{i-1})[\varphi_\mathcal{D}]_{f_{i-}} + \varphi_\mathcal{N}(u_i, u_{i+1})[\varphi_\mathcal{D} - \varphi_\mathcal{A}]_{f_{i+}}, & \text{otherwise.} \end{cases} \tag{4}$$

**Boundary conditions** A means of applying boundary conditions in the model is essential when solving PDEs. Currently available models mostly adopt a convolutional structure for modelling spatiotemporal processes. However, a convolutional structure only allows a constant value to be

padded at the domain boundaries (e.g. zero-padding or mirror-padding), which is only appropriate for the implementation of Dirichlet or periodic boundary condition types. Other types of frequently used boundary conditions are Neumann and Cauchy. They are defined as a derivative of the quantity of interest, and hence cannot be easily implemented in convolutional models. With certain pre-defined boundary condition types, the flux kernels at the boundaries are adjusted accordingly to allow for straightforward boundary condition implementation. For Dirichlet boundary condition, a constant value $u = u_b$ is set as the input $u_{i-1}$ (for the flux kernel $f_{i-}$) or $u_{i+1}$ (for $f_{i+}$) at the corresponding boundary. For Neumann boundary condition $\nu$, the output of the flux kernel $f_{i-}$ or $f_{i+}$ at the corresponding boundary is set to be equal to $\nu$. With Cauchy boundary condition, the solution-dependent derivative is calculated and set as $u_{i-1}$ or $u_{i+1}$ at the corresponding boundary.

**State kernel** The state kernel $\mathcal{S}$ calculates the source/sink and storage terms of Eq. 1. The source/sink (if required) is learned using the module $\Phi_\psi \approx q(u)$, which takes $u$ as input. The storage, $\frac{\partial u}{\partial t}$, is then calculated using the output of the flux kernel and module $\Phi_\psi$ of the state kernel:

$$\mathcal{S}_i = \mathcal{F}_i + \Phi_\psi(u_i) \approx \frac{\partial u_i}{\partial t}. \tag{5}$$

By doing so, the PDE in Eq. 1 is now reduced to a system of coupled ordinary differential equations (ODEs), which are functions of $u_i, u_{i-1}, u_{i+1}$, and $t$. Thus, the solutions of the coupled ODE system can be found using a numerical integration over time. Since first order explicit approaches, such as the Euler method (Butcher, 2008), suffer from numerical instability (Courant et al., 1967; Isaacson & Keller, 1994), we employ the neural ordinary differential equation method (Neural ODE, Chen et al., 2018) to reduce numerical instability via the Runge-Kutta adaptive time-stepping strategy. When using the Neural ODE method, the predicted values of $u$ at time $t$ are fed back into the network as direct inputs to predict $u$ at time $t + 1$. Therefore, the entire training is performed in closed-loop, improving stability and accuracy of the prediction compared to networks trained with teacher forcing, i.e. one-step-ahead prediction (Praditia et al., 2020). The weight update is realized by applying backpropagation through time (indicated by red arrows in Figure 1). In short, FINN takes only the initial condition $u$ at time $t = 0$ and propagates the dynamics forward.

## 4 Experiments, Results & Discussion

### 4.1 Synthetic Dataset

To demonstrate FINN's performance in comparison to other models, four different equations are considered as applications. First, *Burger's equation* (Basdevant et al., 1986) is chosen as a challenging function, as it is a non-linear PDE with $v(u) = u$ that could lead to a shock in the solution $u(x, t)$. Second, the *diffusion-sorption equation* (Nowak & Guthke, 2016) is selected with the non-linear retardation factor $R(u)$ as coefficient for the storage term, which contains a singularity $R(u) \to \infty$ for $u \to 0$ due to the parameter choice. Third, the two-dimensional Fitzhugh-Nagumo equation (Klaasen & Troy, 1984) as candidate for a *diffusion-reaction equation* (Turing, 1952) is selected which is challenging because it consists of two non-linearly coupled PDEs to solve two main unknowns: the activator $u_1$ and the inhibitor $u_2$. Fourth, the Allen-Cahn equation with a cubic reaction term was chosen, leading to multiple jumps in the solution $u(x, t)$. Details on all four equations, data generation and architecture designs can be found in subsection A.4, subsection A.5, subsection A.6, subsection A.7 of the appendix, respectively.

For each problem, three different datasets are generated (by conventional numerical simulation): *train*, used to train the models, in-distribution test (*in-dis-test*), being the train data simulated with a longer time span to test the models' generalization ability (extrapolation), and out-of-distribution test (*out-dis-test*). *Out-dis-test* data are used to test a trained ML model under conditions that are far away from training conditions, not only in terms of querying outputs for unseen inputs. Instead, *out-dis-test* data query outputs with regards to changes *not* captured by the inputs. These are changes that the ML tool per definition cannot be made aware of during training. In this work, they are represented by data generated with different initial or boundary condition, to test the generalization ability of the models outside the training distributions. FINN is trained and compared with both spatiotemporal deep learning models such as TCN, ConvLSTM, DISTANA and physics-aware neural network models such as PINN and PhyDNet. All models are trained with ten different random

Table 1: Comparison of MSE and according standard deviation scores across ten repetitions between different deep learning (above dashed line) and physics-aware neural network (below dashed line) methods on the different equations. Best results are reported in bold.

| Eqn. | Model | Params | Train | In-dis-test | Out-dis-test |
|------|-------|--------|-------|-------------|--------------|
| | | | | Dataset | |
| Burger | TCN | 38 500 | $(1.6 \pm 3.4) \times 10^{-1}$ | $(1.8 \pm 3.1) \times 10^{-1}$ | $(1.6 \pm 3.3) \times 10^{-1}$ |
| | ConvLSTM | 13 200 | $(6.8 \pm 9.9) \times 10^{-2}$ | $(1.2 \pm 1.1) \times 10^{-1}$ | $(7.3 \pm 9.5) \times 10^{-2}$ |
| | DISTANA | 25 126 | $(1.8 \pm 1.0) \times 10^{-4}$ | $(4.0 \pm 3.1) \times 10^{-3}$ | $(1.5 \pm 1.6) \times 10^{-3}$ |
| | PINN | 3 021 | $(5.1 \pm 0.5) \times 10^{-4}$ | $(5.0 \pm 8.4) \times 10^{-3}$ | - |
| | PhyDNet | 37 718 | $(7.2 \pm 2.2) \times 10^{-5}$ | $(1.8 \pm 1.6) \times 10^{-1}$ | $(4.5 \pm 2.5) \times 10^{-2}$ |
| | FINN | **421** | $\mathbf{(2.8 \pm 2.9) \times 10^{-6}}$ | $\mathbf{(2.5 \pm 3.1) \times 10^{-6}}$ | $\mathbf{(2.8 \pm 2.9) \times 10^{-6}}$ |
| Diffusion-sorption | TCN | 3 834 | $(9.7 \pm 13.5) \times 10^{-2}$ | $(1.2 \pm 1.7) \times 10^{-1}$ | $(1.1 \pm 1.4) \times 10^{-1}$ |
| | ConvLSTM | 3 960 | $(3.2 \pm 2.9) \times 10^{-2}$ | $(3.0 \pm 2.3) \times 10^{-2}$ | $(5.8 \pm 4.0) \times 10^{-2}$ |
| | DISTANA | 3 739 | $(4.6 \pm 2.5) \times 10^{-5}$ | $(2.4 \pm 2.5) \times 10^{-3}$ | $(4.6 \pm 4.6) \times 10^{-3}$ |
| | PINN | 3 042 | $(4.7 \pm 8.4) \times 10^{-5}$ | $(4.1 \pm 8.7) \times 10^{-3}$ | - |
| | PhyDNet | 37 815 | $\mathbf{(3.5 \pm 1.7) \times 10^{-5}}$ | $(9.1 \pm 15.4) \times 10^{-3}$ | $(1.7 \pm 0.9) \times 10^{-2}$ |
| | FINN | **528** | $(4.7 \pm 4.9) \times 10^{-5}$ | $\mathbf{(1.3 \pm 1.3) \times 10^{-4}}$ | $\mathbf{(4.1 \pm 4.0) \times 10^{-5}}$ |
| Diffusion-reaction | TCN | 31 734 | $(1.4 \pm 0.9) \times 10^{-2}$ | $(4.7 \pm 2.1) \times 10^{-1}$ | $(1.5 \pm 0.8) \times 10^{-1}$ |
| | ConvLSTM | 24 440 | $(8.7 \pm 21.3) \times 10^{-3}$ | $(9.3 \pm 4.9) \times 10^{-2}$ | $(1.5 \pm 1.3) \times 10^{-2}$ |
| | DISTANA | 75 629 | $(4.0 \pm 3.4) \times 10^{-3}$ | $(1.8 \pm 0.6) \times 10^{-1}$ | $(1.3 \pm 0.9) \times 10^{-2}$ |
| | PINN | 3 062 | $(2.7 \pm 1.8) \times 10^{-4}$ | $(7.0 \pm 6.3) \times 10^{-2}$ | - |
| | PhyDNet | 185 589 | $(7.5 \pm 0.9) \times 10^{-5}$ | $(7.8 \pm 1.8) \times 10^{-2}$ | $(3.5 \pm 1.3) \times 10^{-2}$ |
| | FINN | **882** | $(1.3 \pm 0.3) \times 10^{-4}$ | $\mathbf{(2.1 \pm 0.5) \times 10^{-3}}$ | $\mathbf{(6.1 \pm 0.3) \times 10^{-3}}$ |

seeds using PyTorch's default weight initialization, and mean and standard deviation of the prediction errors are summarized in Table 1 for *train*, *in-dis-test* and *out-dis-test*. The details of each run are reported in the appendix. It is noteworthy that PINN cannot be tested on the *out-dis-test* dataset, since PINN assumes that the unknown variable $u$ is an explicit function of $x$ and $t$, and hence, when the initial or boundary condition is changed, the function will also be different and no longer valid.

**Burger's**   The predictions of the best trained model of each method for the *in-dis-test* and the *out-dis-test* data are shown in Figure 2 and Figure 3, respectively. Both TCN and ConvLSTM fail to produce reasonable predictions, but qualitatively the other models manage to capture the shape of the data sufficiently, even towards the end of the *in-dis-test* period, where closed loop prediction has been applied for 380 time steps (after 20 steps of teacher forcing, see subsection A.4 for details). DISTANA, PINN, and FINN stand out in particular, but FINN produces more consistent and accurate predictions, evidenced by the mean value of the prediction errors. When tested against data generated with a different initial condition (*out-dis-test*), all models except for TCN and PhyDNet

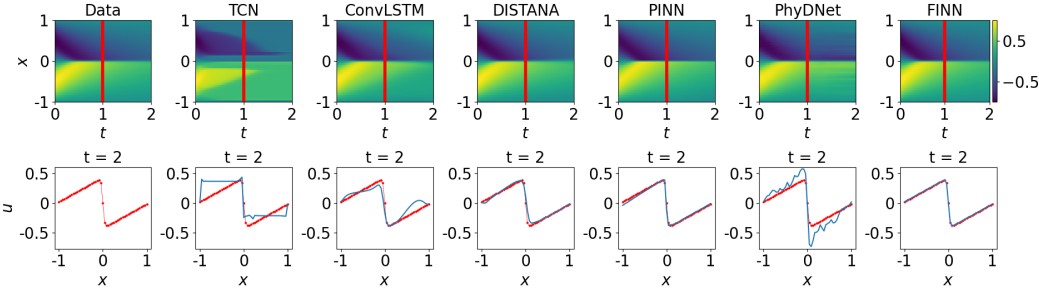

Figure 2: Plots of Burger's data (red) and *in-dis-test* (blue) prediction using different models. The plots in the first row show the solution over $x$ and $t$ (the red lines mark the transition from *train* to *in-dis-test*), the second row visualizes the best model's solution distributed in $x$ at $t = 2$.

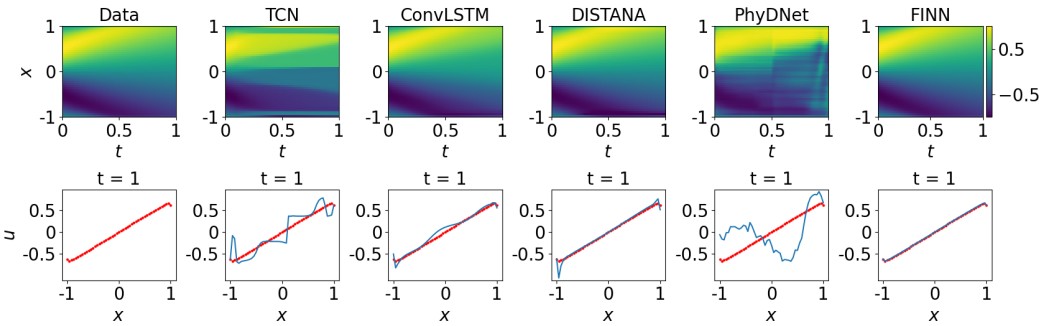

Figure 3: Plots of Burger's data (red) and prediction (blue) of *out-dis-test* data using different models. The plots in the first row show the solution over $x$ and $t$, the second row visualizes the best model's solution over $x$ at $t = 1$.

perform well. However, FINN still outperforms the other models with a significantly lower prediction error. The advective velocity learned by FINN's module $\varphi_{\mathcal{A}}$ is shown in Figure 8 (left) and verifies that it successfully learned the advective velocity to be described by an identity function.

**Diffusion-sorption**   The predictions of the best trained model of each method for the concentration $u$ from the diffusion-sorption equation are shown in Figure 4 and Figure 5. TCN and ConvLSTM are shown to perform poorly even on the *train* data, evidenced by the high mean value of the prediction errors. On *in-dis-test* data, all models successfully produce relatively accurate predictions. However, when tested against different boundary conditions (*out-dis-test*), only FINN is able to capture the modifications and generalize well. The other models are shown to still overfit to the different boundary condition used in the train data (as detailed in subsection A.5). The retardation factor $R(u)$ learned by FINN's $\varphi_{\mathcal{D}}$ module is shown in Figure 8 (second from left). The plot shows that the module learned the Freundlich retardation factor with reasonable accuracy.

**Diffusion-reaction**   The predictions of the best trained model of each method for activator $u_1$ in the diffusion-reaction equation are shown in Figure 6 and Figure 7. TCN is again shown to fail to learn sufficiently from the *train* data. On *in-dis-test* data, DISTANA and the physics-aware models all predict with reasonable accuracy. When tested against data with different initial condition (*out-dis-test*), however, DISTANA and PhyDNet produce predictions with lower accuracy, and we find that FINN is the only model producing relatively low prediction errors. The reaction functions learned by FINN's $\Phi_\psi$ module are shown in Figure 8. The plots show that the module successfully learned the Fitzhugh-Nagumo reaction function, both for the activator and inhibitor.

**Discussion**   Even with a high number of parameters, the prediction errors obtained using the pure ML methods (TCN, ConvLSTM, DISTANA) are worse compared to the physics-aware models, as

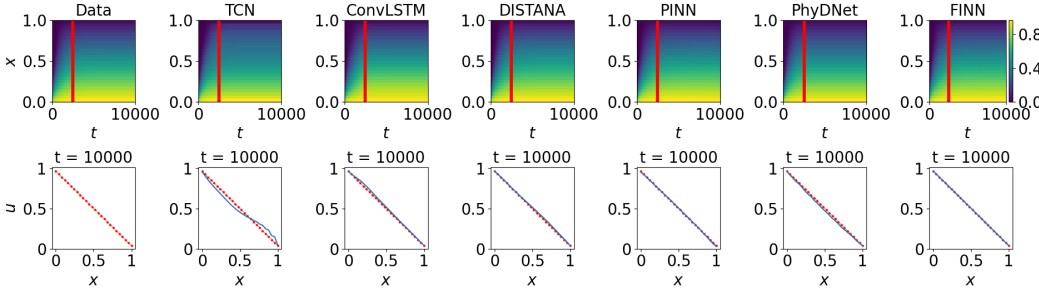

Figure 4: Plots of diffusion-sorption's dissolved concentration (red) and *in-dis-test* prediction (blue) using different models. The first row shows the solution over $x$ and $t$ (red lines mark the transition from *train* to *in-dis-test*), the second row visualizes the solution distributed in $x$ at $t = 10\,000$.

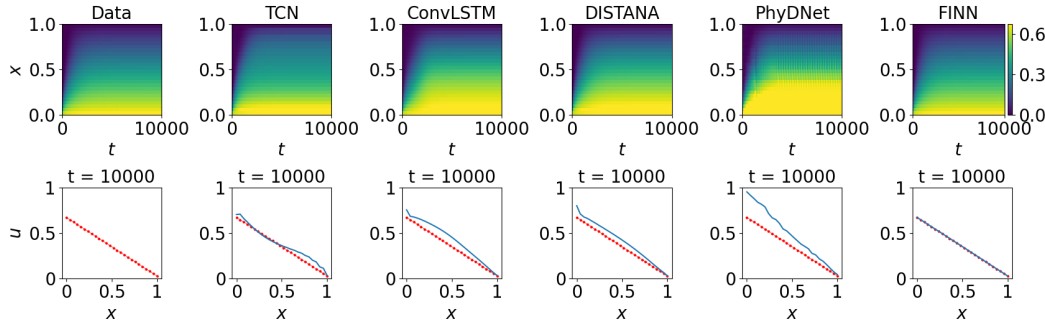

Figure 5: Plots of diffusion-sorption's dissolved concentration data (red) and prediction (blue) of *out-dis-test* data using different models. The plots in the first row show the solution over $x$ and $t$, the second row visualizes the solution distributed in $x$ at $t = 10\,000$.

shown in Table 1. As a physics-aware model, PhyDNet also possesses a high number of parameters. However, most of the parameters are allocated to the data-driven part (i.e. ConvLSTM branch), compared to the physics-aware part (i.e. PhyCell branch). In contrast to the other pure ML methods, DISTANA predicts with higher accuracy. This could act as an evidence that appropriate structural design of a neural network is as essential as physical knowledge to regularize the model training.

Among the physics-aware models, PINN and PhyDNet lie on different extremes. On one side, PINN requires complete knowledge of the modelled system in form of the equation. This means that all the functions, such as the advective velocity in the Burger's equation, the retardation factor in the diffusion-sorption equation, and the reaction function in the diffusion-reaction equation have to be pre-defined in order to train the model. This leaves less room for learning from data and could be error-prone if the designer's assumption is incorrect. On the other side, PhyDNet relies more heavily on the data-driven part and, therefore, could overfit the train data. This can be shown by the fact that PhyDNet reaches the lowest training errors for the diffusion-sorption and diffusion-reaction equation predictions compared to the other models, but its performance significantly deteriorates when applied to *in-* and *out-dis-test* data. Our proposed model, FINN, lies somewhere in the middle, compromising between putting physical knowledge into the model and leaving room for learning from data. As a consequence, FINN shows excellent generalization ability. It significantly outperforms the other models up to multiple orders of magnitude, especially on *out-dis-test* data, when tested with different initial and boundary conditions; which is considered a particularly challenging task for ML models. Furthermore, the structure of FINN allows the extractions of learned functions such as the advective velocity, retardation factor, and reaction functions, showing good interpretability of the model.

We also show that FINN properly handles the provided numerical boundary condition, as evidenced when applied to the test data that is generated with a different left boundary condition value, visual-

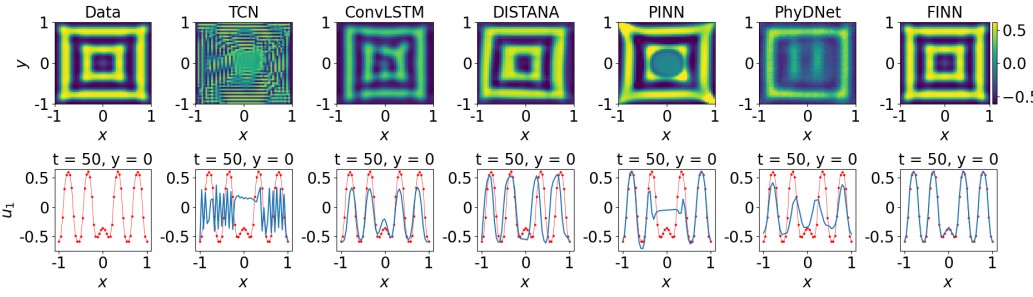

Figure 6: Plots of diffusion-reaction's activator data $u_1$ (red) and extrapolated prediction (blue) using different models. The plots in the first row show the solution distributed over $x$ and $y$ at $t = 50$, and the plots in the second row show the solution distributed in $x$ at $y = 0$ and $t = 50$.

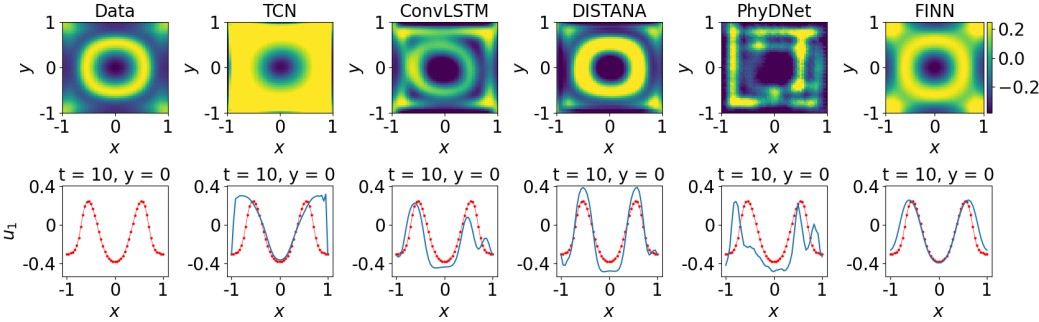

Figure 7: Plots of diffusion-reaction's activator data (red) and prediction (blue) of unseen dataset using different models. The plots in the first row show the solution distributed over $x$ and $y$ at $t = 10$, and the plots in the second row show the solution distributed in $x$ at $y = 0$ and $t = 10$.

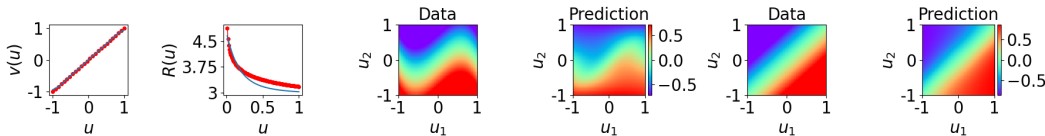

Figure 8: Plots of the learned functions (blue) as a function of $u$ compared to the data (red) for Burger's (left) and diffusion-sorption (second from left). The learned activator $u_1$ and inhibitor $u_2$ reaction functions (right plots of the pairs in center and right) in the diffusion-reaction equation are contrasted to the corresponding ground truth (left plots of the pairs).

ized in Figure 5. Here, the test data is generated with a Dirichlet boundary condition $u(0, t) = 0.7$, which is lower than the value used in the train data, $u(0, t) = 1.0$. However, FINN is the only model that appropriately processes this different boundary condition value so that the prediction fits the test data nicely. The other models overestimate their prediction by consistently showing a tendency to still fit the prediction to the higher boundary condition value encountered during training.

Even though the spatial resolution used for the synthetic data generation is relatively coarse, leading to sparse data availability, FINN generalizes well. PINN, on the other hand, slightly suffers from the low resolution of the train data, although it still shows reasonable performance for the three test cases. Nevertheless, we conducted an experiment showing that PINN performs slightly better and more consistently when trained on higher resolution data (see appendix, subsection A.10), albeit still worse than FINN on coarse data. Therefore, we conclude that FINN is also applicable to real observation data that are often available only in low resolution, and/or in limited quantity. We demonstrate this further in the next section, when we apply FINN to real experimental data. FINN's generalization ability is superior to PINN, due to the fact that it is not possible to apply a trained PINN model to predict data with different initial or boundary condition.

In terms of interpretability, FINN allows the extraction of functions learned by its dedicated modules. These learned functions can be compared with the real functions that generated the data (at least in the synthetic data case); examples are shown in Figure 8. The learned functions are the main data-driven discovery part of FINN and can also be used as a physical plausibility check. PhyDNet also comprises of a physics-aware and a data-driven part. However, it is difficult, if not impossible, to infer the learned equation from the model. Furthermore, the data-driven part of PhyDNet does not possess comparable interpretability, and could lead to overfitting as discussed earlier.

## 4.2 EXPERIMENTAL DATASET

Real observation data are often available in limited amount, and they only provide partial information about the system of interest. To demonstrate how FINN can cope with real observation data, we use experimental data for the diffusion-sorption problem. The experimental data are collected from three different core samples that are taken from the same geographical area: #1,#2, and #2B

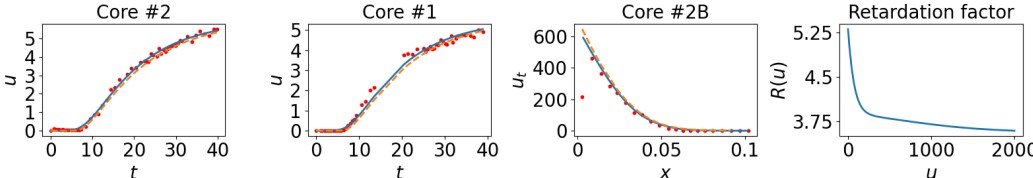

Figure 9: Breakthrough curve prediction of FINN (blue line) during training using data from core sample #2 (left), during testing using data from core sample #1 (second from left) and total concentration profile prediction using data from core sample #2B (second from right). The predictions are compared with the experimental data (red circles) and the results obtained using the physical model (orange dashed line). The learned retardation factor $R(u)$ is shown in the right-most plot.

(see subsection A.12 in the appendix for details). The objective of this experiment is to learn the retardation factor function from one of the core samples that concurrently applies to all the other samples. For this particular purpose, we only implement FINN since the other models have no means of learning the retardation factor explicitly. Here, we use the module $\varphi_{\mathcal{D}}$ to learn the retardation factor function, and we assume that the diffusion coefficient values of all the core samples are known. FINN is trained with the breakthrough curve of $u$, which is the dissolved concentration only at $x = L|_{0 \leq t \leq t_{\text{end}}}$ (i.e. only 55 data points).

FINN reaches a higher accuracy for the training with core sample #2, with MSE $= 4.84 \times 10^{-4}$ compared to a calibrated physical model from Nowak & Guthke (2016) with MSE $= 1.06 \times 10^{-3}$, because the latter has to assume a specific function $R(u)$ with a few parameters for calibration. Our learned retardation factor is then applied and tested to core samples #1 and #2B. Figure 9 shows that FINN's prediction accuracy is competitive compared to the calibrated physical model. For core sample #1, FINN's prediction has an accuracy of $1.37 \times 10^{-3}$ compared to the physical model that underestimates the breakthough curve (i.e. concentration profile) with MSE $= 2.50 \times 10^{-3}$. Core sample #2B has significantly longer length than the other samples, and therefore a no-flow Neumann boundary condition was assumed at the bottom of the core. Because there is no breakthrough curve data available for this specific sample, we compare the prediction against the so-called total concentration profile $u(x, t_{\text{end}})$ at the end of the experiment. FINN produced a prediction with an accuracy of $1.16 \times 10^{-3}$, whereas the physical model overestimates the concentration with MSE $= 2.73 \times 10^{-3}$. To briefly summarize, FINN is able to learn the retardation factor from a sparse experimental dataset and apply it to other core samples with similar soil properties with reasonable accuracy, even applying to a different boundary condition type.

## 5    CONCLUSION

Spatiotemporal dynamics often can be described by means of advection-diffusion type equations, such as Burger's, diffusion-sorption, or diffusion-reaction equations. When modeling those dynamics with ANNs, large efforts must be taken to prevent the models from overfitting (given the model is able to learn the problem at all). The incorporation of physical knowledge can be seen as regularization, yielding robust predictions beyond the training data distribution.

With FINN, we have introduced a modular, physics-aware neural network with excellent generalization abilities beyond different initial and boundary conditions, when contrasted to pure ML models and other physics-aware models. FINN is able to model and extract unknown constituents of differential equations, allowing high interpretability and an assessment of the plausibility of the model's out-of-distribution predictions. As next steps we seek to apply FINN beyond second order spatial derivatives, improve its scalability to large datasets, and make it applicable to heterogeneously distributed data (i.e. represented as graphs) by modifying the module $\varphi_{\mathcal{N}}$ to approximate variable and location-specific stencils (for more details on limitations of FINN, the reader is referred to subsection A.2). Another promising future direction is the application of FINN to real-world weather data.

AUTHOR CONTRIBUTIONS

Hidden for anonymization reasons.

ACKNOWLEDGMENTS

We thank the reviewers for suggesting the evaluation of polynomial fitting and least squares regression as additional baselines as well as modeling another challenging equation (we chose Allen-Cahn with its cubic term) to demonstrate the wide applicability of the method, which we have added to the appendix. The results confirm our claims and further suggestions helped us to increase the comprehensibility and clarity of the method. Additional parts are hidden for anonymization reasons

REPRODUCIBILITY STATEMENT

Code and data that are used for this paper can be found in the repository \*\*\*\*\* *[hidden for anonymization reasons. Instead, the supplementary material of this submission contains an anonymous version of the repository's* README.md *file along with the according data and model scripts.]*

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

## A  APPENDIX

This appendix is structured as follows: The detailed differences with the benchmark models used in this work are presented in subsection A.1. Then, limitations of FINN are discussed briefly in subsection A.2. Detailed numerical derivation of the first and second order spatial derivative is shown in subsection A.3. Additional details on the equations used as well as model specifications and in-depth results are presented in subsection A.4 (Burger's), subsection A.5 (diffusion-sorption), subsection A.6 (diffusion-reaction), and subsection A.7 (Allen-Cahn). Additional polynomial-fitting baselines to account for the equations are reported in subsection A.8, including an ablation where we replace the neural network modules of FINN by polynomials. A robustness analysis of FINN can be found in subsection A.9, where our method is evaluated on noisy data from all equations. Results of training PINN with high-resolution data are reported in subsection A.10, whereas results of the original PhyDNet architecture are outlined in subsection A.11. The soil parameters and simulation domain used in the diffusion-sorption laboratory experiment are presented in subsection A.12.

### A.1  DISTINCTION TO RELATED WORK

**Pure ML models**  Originally, FINN was inspired by DISTANA, a pure ML model proposed by Karlbauer et al. (2019). While DISTANA has large similarities to Shi et al. (2015)'s ConvLSTM, it propagates lateral information through an additional latent map—instead of reading lateral information from the input map directly, as done in ConvLSTM—and transforms that lateral information by means of a user-defined combination of arbitrary layers. Additionally, the processing of the two-point flux approximation in FINN using the lateral information, is more akin to DISTANA, albeit motivated and augmented by physical knowledge.

Accordingly, the lateral information flow is guaranteed to behave in a physically plausible manner. That is, quantity can either be locally generated by a source (increased), locally absorbed by a sink (decreased), or spatially distributed (move to neighboring cells). Terminology separates the spatially distribution of quantity into diffusion and advection. Diffusion describes the equalization of quantity from high to low concentration levels, whereas advection is defined as the bulk motion of a large group of particles/atoms caused by external forces. FINN ensures the conservation of laterally propagating quantity, i.e. what flows from left to right will effectively cause a decrease left and an increase right. Pure ML models (without physical constraints) cannot be guaranteed to adhere to these fundamental rules.

**PINN**  Since ML models guided by physical knowledge not only behave empirically plausible but also require fewer parameters and generalize to much broader extent, numerous approaches have been proposed recently to combine artificial neural networks with physical knowledge. Raissi et al. (2019) introduced the physics-informed neural network (PINN), a concrete and outstanding model that explicitly learns a provided PDE. As a result, PINN mimics e.g. Burger's equation (see equation 7) by learning the quantity function $u(x, t)$ for defined initial and boundary conditions with a feedforward network. The partial derivatives are implicitly realized by automatic differentiation of the feedforward network (representing $u$) with respect to the input variable $x$ or $t$. The learned neural network thus satisfies the constraints defined by the partial derivatives.

This method has the advantage that it explicitly provides a solution of the desired function $u(x, t)$ and, correspondingly, predictions can be generated for an arbitrary combination of input values, circumventing the need for simulating the entire domain with e.g. a carefully chosen simulation step size. In contrast, FINN does not learn the explicit function $u(x, t)$ defined for particular initial and boundary conditions, but approximates the distinct components of the function. These components are combined as suggested by the physical equation to result in a compositional model that is more universally applicable. That is, in stark contrast to PINN, the compositional function learned

by FINN can be applied to varying initial and boundary conditions, since the learned individual components provide the same functionality as the corresponding components of the PDE when processed by a numerical solver. Applying the numerical solver, i.e. the finite volume method (FVM), however, requires either complete knowledge of the equation or careful calibration of the unknowns by choosing equations from a library of possible solutions and tuning the parameters. FINN's data driven component can reveal unknown relations, such as the retardation factor of a function, without the need for subjective prior assumptions.

**Learning derivatives via convolution** An alternative and much addressed approach of implanting physical plausibilty into artificial neural networks is to implicitly learn the first $n$ derivatives using appropriately designed convolution kernels (Long et al., 2018; Li et al., 2020a;b; Guen & Thome, 2020; Yin et al., 2020; Sitzmann et al., 2020). These methods exploit the link that most PDEs, such as $u(t, x, y, \dots)$, can be reformulated as

$$\frac{\partial u}{\partial t} = F\left(t, x, y, \dots, u, \frac{\partial u}{\partial x}, \frac{\partial u}{y}, \frac{\partial u}{\partial x \partial y}, \frac{\partial^2 u}{\partial x^2}, \dots\right). \tag{6}$$

When learning partial derivatives up to order $n$ and setting irrelevant features to zero, these methods have, in principle, the capacity to represent most PDEs. However, the degrees of freedom in these methods are still very high and can fail to safely guide the ML algorithm. FINN accounts for the first and second derivative by learning an according stencil in the module $\varphi_\mathcal{N}$ and combining this stencil with the case-sensitive ReLU module $\mathcal{R}$, which allows a precise control of the information flow resulting from the first and second derivative. More importantly, convolutional structure only allows implementation of Dirichlet and periodic boundary condition (by means of zero- or mirror-padding), and is not appropriate for implementation of other boundary condition types.

**Learning ODE coefficients** The group around Brenner and Hoyer (Bar-Sinai et al., 2019; Kochkov et al., 2021; Zhuang et al., 2021) follow another line of research about learning the coefficients of ordinary differential equations (ODEs); note that PDEs can be transformed into a set of coupled ODEs by means of polynomial expansion or spectral differentiation as shown in Bar-Sinai et al. (2019). The physical constraint in these works is mostly realized by using the temporal derivative as loss.

While this approach share similarities with our method by incorporating physically motivated inductive bias into the model, it uses this bias mainly to improve interpolation from coarse to finer grid resolutions and thus to accelerate simulation. Our work focuses on discovering unknown relationships/laws (or re-discovering laws in the case of the synthetic examples), such as the advective velocity in the Burger's example, the retardation factor function in the diffusion-sorption example, and the reaction function in the diffusion-reaction example. Additionally, the works by Brenner and Hoyer employ a convolutional structure, which is only applicable to Dirichlet or periodic boundary conditions, and it suffers from a slight instability during training when the training data trajectory is unrolled for a longer period. In contrast, FINN employs the flux kernel, calculated at all control volume surfaces, which enables the implementation and discovery of various boundary conditions. Furthermore and in contrast to Brenner and Hoyer, FINN employs the Neural ODE method as the time integrator to reduce numerical instability during training with long time series. However and in accord with this line of research, FINN is also able to generalize well when trained with a relatively sparse dataset (coarse resolution), reducing the computational burden.

**Numerical ODE solvers** Traditional numerical solvers for ordinary differential equations (ODEs) can be seen as the pure-physics contrast to FINN. However, these do not have a learning capacity to reveal unknown dependencies or functions from data. Also, as shown in (Yin et al., 2020), a pure Neural ODE without physical inductive bias does not reach the same level of accuracy as physics-aware neural network models. This is underlined by our field experiment where FINN reached lower errors on a real-world dataset compared to a conventional FVM model.

## A.2 Limitations of FINN

While the largest limitation of our current method can be seen in the capacity to only represent first and second order spatial derivatives, this is an issue that we will address in follow up work. Still, FINN can already be applied to a very wide range of problems as most equations in fact only

depend on up to the second spatial derivative. So far, FINN is only applicable to homogeneously distributed data—we intend to extend it to heterogeneous data from graphs. Although we have successfully applied FINN to 2D diffusion reaction data, the training time is considerable. While, according to our observations, this appears to be a common issue for physics-aware neural networks, the implementation of custom-convolution layers could widen this bottleneck with today's hardware-accelerated computation of convolution operations.

### A.3 LEARNING THE NUMERICAL STENCIL

Semantically, the $\varphi_\mathcal{N}$ module learns the numerical stencil, that is the geometrical arrangement of a group of neighboring cells to approximate the derivative numerically, effectively learning the first spatial derivative $\frac{\partial u}{\partial x}$ from both $[u_{i-1}, u_i]$ and $[u_i, u_{i+1}]$, which are the inputs to the $\varphi_{\mathcal{N}_{i-}}$ and $\varphi_{\mathcal{N}_{i+}}$ module, respectively.

The lateral information flowing from $u_{i-1}$ and $u_{i+1}$ toward $u_i$ is controlled by the $\varphi_\mathcal{A}$ (advective flux, i.e. bulk motion of many particles/atoms that can either move to the left or to the right) and the $\varphi_\mathcal{D}$ (diffusive flux, i.e. drive of particles/atoms to equilibrium from regions of high to low concentration) modules. Since the advective flux can only move either to the left or to the right, it will be considered only in the left flux kernel ($f_{i-}$) or in the right flux kernel ($f_{i+}$), and not both at the same time. The case-sensitive ReLU module $\mathcal{R}$ (Eq. 3) decides on this, by setting the advective flux in the irrelevant flux kernel to zero (effectively depending on the sign of the output of $\varphi_\mathcal{A}$). Thus, the advective flux is only considered from either $u_{i-1}$ or $u_{i+1}$ to $u_i$, which amounts to the first order spatial derivative.

The diffusive flux, on the other hand, can propagate from both sides towards the control volume of interest $u_i$ and, hence, the second order spatial derivative, accounting for the difference between $u_{i-1}$ and $u_{i+1}$, has to be applied. In our method, this is realized through the combination of the $\varphi_\mathcal{N}$ and $\varphi_\mathcal{D}$ modules, calculating the diffusive fluxes $\delta_- = \varphi_\mathcal{N}(u_{i-1}, u_i)\varphi_\mathcal{D}(u_i)$ and $\delta_+ = \varphi_\mathcal{N}(u_i, u_{i+1})\varphi_\mathcal{D}(u_i)$ inside of the respective left and right flux kernel. The combination of these two deltas in the state kernel ensures the consideration of the diffusive fluxes from left (including $u_{i-1}$) and from right (including $u_{i+1}$), resulting in the ability to account for the second order spatial derivative.

Technically, the first, i.e. $[-1, 1]$, and second, i.e. $[1, -2, 1]$, order spatial differentiation schemes are common definitions and a derivation can be found, for example, in Fornberg (1988). However, a quick derivation of the Laplace scheme $[1, -2, 1]$ can be formulated as follows. Define the second order spatial derivative as

$$\frac{\partial^2 u}{\partial x^2} \approx \frac{(\partial u/\partial x)|_{i-} - (\partial u/\partial x)|_{i+}}{\Delta x}$$

with $(\partial u/\partial x)|_{i-} \approx (u_{i-1} - u_i)/\Delta x$ and $(\partial u/\partial x)|_{i+} \approx (u_i - u_{i+1})/\Delta x$. Then

$$\frac{\partial^2 u}{\partial x^2} \approx \frac{(u_{i-1} - u_i) - (u_i - u_{i+1})}{\Delta x^2} = \frac{u_{i-1} - 2u_i + u_{i+1}}{\Delta x^2}$$

hence the $[+1, -2, +1]$ as coefficients.

### A.4 BURGER

The Burger's equation is commonly employed in various research areas, including fluid mechanics.

**Data** The 1D generalized Burger's equation is written as

$$\frac{\partial u}{\partial t} = -v(u)\frac{\partial u}{\partial x} + D\frac{\partial^2 u}{\partial x^2}, \tag{7}$$

where the main unknown is $u$, the advective velocity is denoted as $v(u)$ which is a function of $u$ and the diffusion coefficient is $D = 0.01/\pi$. In the current work, the advective velocity function is chosen to be an identity function $v(u) = u$ to reproduce the experiment conducted in the PINN paper (Raissi et al., 2019).

The simulation domain for the ***train* data** is defined with $x = [-1, 1]$, $t = [0, 1]$ and is discretized with $N_x = 49$ spatial locations, and $N_t = 201$ simulation steps. The initial condition is defined as $u(x, 0) = -\sin(\pi x)$, and the boundary condition is defined as $u(-1, t) = u(1, t) = 0$.

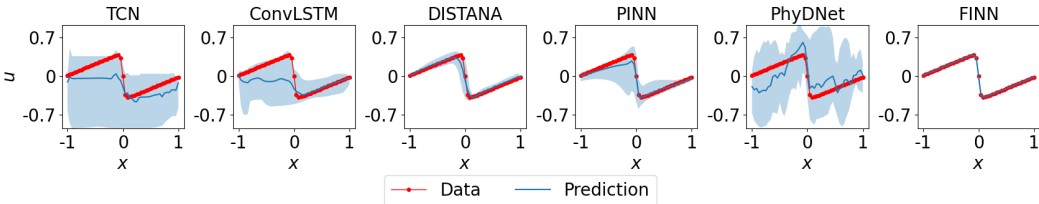

Figure 10: Prediction mean over ten different trained models (with 95% confidence interval) of the Burger's equation at $t = 2$ for the *in-dis-test* dataset.

***In-dis-test* data** is simulated with $x = [-1, 1]$ and a time span of $t = [1, 2]$ and $N_t = 401$. Initial condition is taken from the train data at $t = 1$ and boundary conditions are also similar to the train data.

The simulation domain for the ***out-dis-test* data** is identical with the train data, except for the initial condition that is defined as $u(x, 0) = \sin(\pi x)$.

**Model architectures** Both **TCN** and **ConvLSTM** are designed to have one input neuron, one hidden layer of size 32, and one output neuron. The lateral and dynamic input- and output sizes of the **DISTANA** model are set to one and a hidden layer of size 32 is used. The pure ML models were trained on the first 150 time steps and validated on the remaining 50 time steps of the train data (applying early stopping). Also, to prevent the pure ML models from diverging too much in closed loop, the boundary data are fed into the models as done during teacher forcing. **PINN** was defined as a feedforward network with the size of [2, 20, 20, 20, 20, 20, 20, 20, 20, 1] (8 hidden layers, each contains 20 hidden neurons), as reported in the original work by Raissi et al. (2019). **PhyDNet** was defined with the PhyCell containing 32 input dimensions, 7 hidden dimensions, 1 hidden layer, and the ConvLSTM containing 32 input dimensions, 32 hidden dimensions, 1 hidden layer. For **FINN**, the modules $\varphi_{\mathcal{N}}$, $\varphi_{\mathcal{D}}$, $\mathcal{R}$ and $\varphi_{\mathcal{A}}$ were used, with $\varphi_{\mathcal{A}}$ defined as a feedforward network with the size of [1, 10, 20, 10, 1] that takes $u$ as an input and outputs the advective velocity $v(u)$, and $\varphi_{\mathcal{D}}$ as a learnable scalar that learns the diffusion coefficient $D$. All models are trained until convergence using the L-BFGS optimizer, except for PhyDNet, which is trained with the Adam optimizer and a learning rate of $1 \times 10^{-3}$ due to stability issues when training with the L-BFGS optimizer.

**Additional results** Individual errors of the ten different training runs as well as visualizations for the *train* (Table 2), *in-dis-test* (Table 3 and Figure 10), and *out-dis-test* (Table 4 and Figure 11) datasets.

Table 2: Closed loop MSE on the *train* data from ten different training runs for each model for the Burger's equation.

| Run | TCN | ConvLSTM | DISTANA | PINN | PhyDNet | FINN |
|-----|-----|----------|---------|------|---------|------|
| 1 | $4.1 \times 10^{-2}$ | $7.9 \times 10^{-2}$ | $1.2 \times 10^{-4}$ | $6.1 \times 10^{-4}$ | $8.0 \times 10^{-5}$ | $1.9 \times 10^{-6}$ |
| 2 | $5.1 \times 10^{-2}$ | $1.7 \times 10^{-1}$ | $1.8 \times 10^{-4}$ | $4.1 \times 10^{-4}$ | $4.7 \times 10^{-5}$ | $1.5 \times 10^{-6}$ |
| 3 | $3.6 \times 10^{-2}$ | $1.3 \times 10^{-2}$ | $6.4 \times 10^{-5}$ | $4.9 \times 10^{-4}$ | $6.8 \times 10^{-5}$ | $2.2 \times 10^{-6}$ |
| 4 | $4.3 \times 10^{-2}$ | $2.7 \times 10^{-4}$ | $2.9 \times 10^{-4}$ | $5.6 \times 10^{-4}$ | $1.2 \times 10^{-4}$ | $2.5 \times 10^{-6}$ |
| 5 | $5.4 \times 10^{-2}$ | $2.1 \times 10^{-4}$ | $2.2 \times 10^{-4}$ | $5.1 \times 10^{-4}$ | $6.7 \times 10^{-5}$ | $1.6 \times 10^{-6}$ |
| 6 | $3.9 \times 10^{-2}$ | $1.5 \times 10^{-3}$ | $1.3 \times 10^{-4}$ | $5.2 \times 10^{-4}$ | $5.5 \times 10^{-5}$ | $9.9 \times 10^{-7}$ |
| 7 | $6.6 \times 10^{-2}$ | $4.1 \times 10^{-4}$ | $5.4 \times 10^{-5}$ | $5.2 \times 10^{-4}$ | $5.1 \times 10^{-5}$ | $2.2 \times 10^{-6}$ |
| 8 | $1.2 \times 10^{0}$ | $3.6 \times 10^{-2}$ | $2.7 \times 10^{-4}$ | $4.8 \times 10^{-4}$ | $5.6 \times 10^{-5}$ | $3.0 \times 10^{-6}$ |
| 9 | $3.4 \times 10^{-2}$ | $5.5 \times 10^{-2}$ | $3.9 \times 10^{-4}$ | $5.5 \times 10^{-4}$ | $9.9 \times 10^{-5}$ | $1.1 \times 10^{-5}$ |
| 10 | $6.8 \times 10^{-2}$ | $3.2 \times 10^{-1}$ | $1.1 \times 10^{-4}$ | $5.0 \times 10^{-4}$ | $8.1 \times 10^{-5}$ | $4.5 \times 10^{-7}$ |

Table 3: Closed loop MSE on *in-dis-test* data from ten different training runs for each model for the Burger's equation.

| Run | TCN | ConvLSTM | DISTANA | PINN | PhyDNet | FINN |
|---|---|---|---|---|---|---|
| 1 | $3.0 \times 10^{-2}$ | $1.2 \times 10^{-1}$ | $2.6 \times 10^{-3}$ | $4.1 \times 10^{-3}$ | $1.1 \times 10^{-1}$ | $1.6 \times 10^{-6}$ |
| 2 | $7.0 \times 10^{-2}$ | $3.7 \times 10^{-1}$ | $1.1 \times 10^{-3}$ | $1.3 \times 10^{-3}$ | $2.5 \times 10^{-1}$ | $1.3 \times 10^{-6}$ |
| 3 | $2.4 \times 10^{-2}$ | $2.0 \times 10^{-1}$ | $8.9 \times 10^{-4}$ | $2.9 \times 10^{-3}$ | $3.4 \times 10^{-2}$ | $1.9 \times 10^{-6}$ |
| 4 | $5.1 \times 10^{-2}$ | $5.6 \times 10^{-3}$ | $6.2 \times 10^{-3}$ | $3.7 \times 10^{-3}$ | $4.7 \times 10^{-1}$ | $2.1 \times 10^{-6}$ |
| 5 | $5.4 \times 10^{-2}$ | $2.6 \times 10^{-3}$ | $2.6 \times 10^{-3}$ | $2.4 \times 10^{-3}$ | $1.1 \times 10^{-2}$ | $1.3 \times 10^{-6}$ |
| 6 | $3.2 \times 10^{-2}$ | $1.0 \times 10^{-2}$ | $1.2 \times 10^{-2}$ | $1.9 \times 10^{-3}$ | $3.2 \times 10^{-1}$ | $6.8 \times 10^{-7}$ |
| 7 | $8.8 \times 10^{-2}$ | $5.5 \times 10^{-3}$ | $1.1 \times 10^{-3}$ | $2.1 \times 10^{-3}$ | $3.3 \times 10^{-1}$ | $1.8 \times 10^{-6}$ |
| 8 | $1.1 \times 10^{0}$ | $1.6 \times 10^{-1}$ | $4.0 \times 10^{-3}$ | $4.0 \times 10^{-4}$ | $2.7 \times 10^{-1}$ | $2.4 \times 10^{-6}$ |
| 9 | $3.3 \times 10^{-2}$ | $1.9 \times 10^{-1}$ | $4.0 \times 10^{-3}$ | $3.0 \times 10^{-2}$ | $2.6 \times 10^{-2}$ | $1.2 \times 10^{-5}$ |
| 10 | $3.6 \times 10^{-1}$ | $9.8 \times 10^{-2}$ | $6.0 \times 10^{-3}$ | $1.5 \times 10^{-3}$ | $8.0 \times 10^{-3}$ | $2.8 \times 10^{-7}$ |

Table 4: Closed loop MSE on *out-dis-test* data from ten different training runs for each model for the Burger's equation.

| Run | TCN | ConvLSTM | DISTANA | PINN | PhyDNet | FINN |
|---|---|---|---|---|---|---|
| 1 | $3.6 \times 10^{-2}$ | $7.2 \times 10^{-2}$ | $5.9 \times 10^{-3}$ | - | $5.3 \times 10^{-2}$ | $1.9 \times 10^{-6}$ |
| 2 | $3.4 \times 10^{-2}$ | $1.9 \times 10^{-1}$ | $4.4 \times 10^{-4}$ | - | $2.1 \times 10^{-2}$ | $1.5 \times 10^{-6}$ |
| 3 | $3.3 \times 10^{-2}$ | $2.9 \times 10^{-3}$ | $2.9 \times 10^{-3}$ | - | $4.5 \times 10^{-2}$ | $2.2 \times 10^{-6}$ |
| 4 | $3.7 \times 10^{-2}$ | $8.9 \times 10^{-4}$ | $1.4 \times 10^{-3}$ | - | $2.0 \times 10^{-2}$ | $2.5 \times 10^{-6}$ |
| 5 | $4.2 \times 10^{-2}$ | $1.2 \times 10^{-3}$ | $1.0 \times 10^{-3}$ | - | $3.1 \times 10^{-2}$ | $1.6 \times 10^{-6}$ |
| 6 | $2.7 \times 10^{-2}$ | $1.4 \times 10^{-3}$ | $3.0 \times 10^{-4}$ | - | $1.1 \times 10^{-1}$ | $9.9 \times 10^{-7}$ |
| 7 | $6.7 \times 10^{-2}$ | $3.3 \times 10^{-2}$ | $6.2 \times 10^{-4}$ | - | $2.8 \times 10^{-2}$ | $2.2 \times 10^{-6}$ |
| 8 | $1.1 \times 10^{0}$ | $4.0 \times 10^{-2}$ | $7.0 \times 10^{-4}$ | - | $4.7 \times 10^{-2}$ | $3.0 \times 10^{-6}$ |
| 9 | $3.0 \times 10^{-2}$ | $8.5 \times 10^{-2}$ | $7.4 \times 10^{-4}$ | - | $4.5 \times 10^{-2}$ | $1.1 \times 10^{-5}$ |
| 10 | $1.2 \times 10^{-1}$ | $3.1 \times 10^{-1}$ | $7.0 \times 10^{-4}$ | - | $4.5 \times 10^{-2}$ | $4.5 \times 10^{-7}$ |

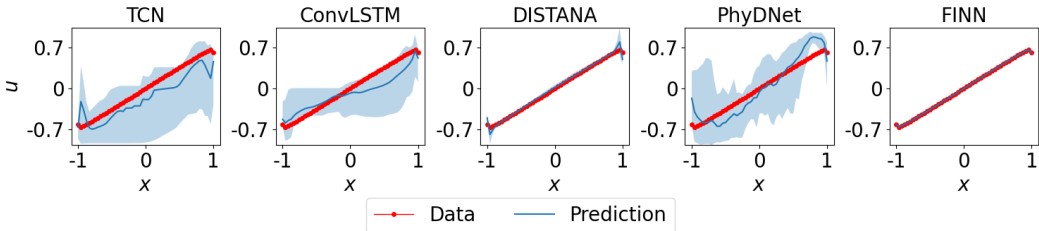

Figure 11: Prediction mean over ten different trained models (with 95% confidence interval) of the Burger's equation at $t = 1$ for the *out-dis-test* data.

A.5 DIFFUSION-SORPTION

The diffusion-sorption equation is another widely applied equation in fluid mechanics. A practical example of the equation is to model contaminant transport in groundwater. Its retardation factor $R$ can be modelled using different closed parametric relations known as sorption isotherms that should be calibrated to observation data.

**Data** The 1D diffusion-sorption equation is written as the following coupled system of equations:

$$\frac{\partial u}{\partial t} = \frac{D}{R(u)} \frac{\partial^2 u}{\partial x^2}, \qquad (8) \qquad\qquad \frac{\partial u_t}{\partial t} = D\phi \frac{\partial^2 u}{\partial x^2}, \qquad (9)$$

where the dissolved concentration $u$ and total concentration $u_t$ are the main unknowns. The effective diffusion coefficient is denoted by $D = 5 \times 10^{-4}$, the retardation factor is $R(u)$, a function of $u$, and the porosity is denoted by $\phi = 0.29$.

In this work, the Freundlich sorption isotherm, see details in (Nowak & Guthke, 2016), was chosen to define the retardation factor:

$$R(u) = 1 + \frac{1 - \phi}{\phi} \rho_s k n_f u^{n_f - 1}, \qquad (10)$$

where $\rho_s = 2\,880$ is the bulk density, $k = 3.5 \times 10^{-4}$ is the Freundlich's parameter, and $n_f = 0.874$ is the Freundlich's exponent.

The simulation domain for the ***train data*** is defined with $x = [0, 1]$, $t = [0, 2\,500]$ and is discretized with $N_x = 26$ spatial locations, and $N_t = 501$ simulation steps. The initial condition is defined as $u(x, 0) = 0$, and the boundary condition is defined as $u(0, t) = 1.0$ and $u(1, t) = D\frac{\partial u}{\partial x}$.

The ***in-dis-test*** data was simulated with $x = [0, 1]$ and with the time span of $t = [2\,500, 10\,000]$ and $N_t = 2\,001$. Initial condition is taken from the train data at $t = 2\,500$ and boundary conditions are also identical to the train data.

The simulation domain for the ***out-dis-test*** data was identical with the *in-dis-test* data, except for the boundary condition that was defined as $u(0, t) = 0.7$.

**Model architectures** **TCN** is designed to have two input neurons, four hidden layers of size [4, 8, 8, 8], and two output neurons. **ConvLSTM** has two input- and output neurons and one hidden layer with 16 neurons. The lateral and dynamic input- and output sizes of the **DISTANA** model are set to one and two, respectively, while a hidden layer of size 12 is used. The pure ML models were trained on the first 400 time steps and validated on the remaining 100 time steps of the train data (applying early stopping). Also, to prevent the pure ML models from diverging too much in closed loop, the boundary data are fed into the models as done during teacher forcing. **PINN** was

Table 5: Closed loop MSE on the *train* data from ten different training runs for each model for the diffusion-sorption equation.

| Run | TCN | ConvLSTM | DISTANA | PINN | PhyDNet | FINN |
|-----|-----|----------|---------|------|---------|------|
| 1 | $3.6 \times 10^{-2}$ | $4.8 \times 10^{-5}$ | $2.1 \times 10^{-5}$ | $2.2 \times 10^{-4}$ | $3.1 \times 10^{-5}$ | $1.1 \times 10^{-4}$ |
| 2 | $2.0 \times 10^{-1}$ | $2.3 \times 10^{-2}$ | $1.6 \times 10^{-5}$ | $2.3 \times 10^{-6}$ | $2.5 \times 10^{-5}$ | $1.4 \times 10^{-5}$ |
| 3 | $8.1 \times 10^{-2}$ | $4.4 \times 10^{-2}$ | $2.7 \times 10^{-5}$ | $1.2 \times 10^{-6}$ | $3.2 \times 10^{-5}$ | $1.1 \times 10^{-4}$ |
| 4 | $2.0 \times 10^{-1}$ | $1.0 \times 10^{-2}$ | $9.3 \times 10^{-5}$ | $2.1 \times 10^{-4}$ | $2.9 \times 10^{-5}$ | $4.1 \times 10^{-5}$ |
| 5 | $1.8 \times 10^{-3}$ | $8.7 \times 10^{-2}$ | $6.2 \times 10^{-5}$ | $2.5 \times 10^{-6}$ | $7.7 \times 10^{-5}$ | $5.3 \times 10^{-6}$ |
| 6 | $2.7 \times 10^{-4}$ | $5.9 \times 10^{-2}$ | $4.0 \times 10^{-5}$ | $1.7 \times 10^{-6}$ | $4.4 \times 10^{-5}$ | $1.9 \times 10^{-5}$ |
| 7 | $3.2 \times 10^{-4}$ | $6.4 \times 10^{-2}$ | $7.7 \times 10^{-5}$ | $8.6 \times 10^{-6}$ | $1.8 \times 10^{-5}$ | $1.5 \times 10^{-5}$ |
| 8 | $1.7 \times 10^{-2}$ | $1.4 \times 10^{-4}$ | $2.5 \times 10^{-5}$ | $2.0 \times 10^{-6}$ | $2.4 \times 10^{-5}$ | $1.2 \times 10^{-4}$ |
| 9 | $4.3 \times 10^{-1}$ | $2.2 \times 10^{-3}$ | $5.9 \times 10^{-5}$ | $9.0 \times 10^{-6}$ | $2.1 \times 10^{-5}$ | $1.2 \times 10^{-5}$ |
| 10 | $6.6 \times 10^{-4}$ | $2.7 \times 10^{-2}$ | $3.9 \times 10^{-5}$ | $1.1 \times 10^{-5}$ | $5.2 \times 10^{-5}$ | $1.7 \times 10^{-5}$ |

Table 6: Closed loop MSE on *in-dis-test* data from ten different training runs for each model for the diffusion-sorption equation.

| Run | TCN | ConvLSTM | DISTANA | PINN | PhyDNet | FINN |
|-----|-----|----------|---------|------|---------|------|
| 1 | $1.9 \times 10^{-2}$ | $7.7 \times 10^{-3}$ | $6.9 \times 10^{-4}$ | $3.0 \times 10^{-2}$ | $3.8 \times 10^{-3}$ | $3.2 \times 10^{-4}$ |
| 2 | $2.4 \times 10^{-1}$ | $1.0 \times 10^{-2}$ | $8.5 \times 10^{-5}$ | $8.6 \times 10^{-4}$ | $8.4 \times 10^{-3}$ | $3.5 \times 10^{-5}$ |
| 3 | $4.7 \times 10^{-2}$ | $2.1 \times 10^{-2}$ | $1.7 \times 10^{-3}$ | $9.3 \times 10^{-5}$ | $2.9 \times 10^{-4}$ | $3.2 \times 10^{-4}$ |
| 4 | $2.4 \times 10^{-1}$ | $2.8 \times 10^{-2}$ | $3.5 \times 10^{-3}$ | $5.2 \times 10^{-3}$ | $3.6 \times 10^{-4}$ | $1.1 \times 10^{-4}$ |
| 5 | $3.5 \times 10^{-2}$ | $7.9 \times 10^{-2}$ | $7.3 \times 10^{-3}$ | $7.9 \times 10^{-6}$ | $2.8 \times 10^{-2}$ | $9.8 \times 10^{-6}$ |
| 6 | $2.4 \times 10^{-2}$ | $5.4 \times 10^{-2}$ | $3.1 \times 10^{-4}$ | $1.4 \times 10^{-5}$ | $4.8 \times 10^{-2}$ | $4.9 \times 10^{-5}$ |
| 7 | $1.4 \times 10^{-3}$ | $4.2 \times 10^{-2}$ | $6.5 \times 10^{-3}$ | $1.8 \times 10^{-4}$ | $1.0 \times 10^{-3}$ | $3.7 \times 10^{-5}$ |
| 8 | $3.9 \times 10^{-2}$ | $2.2 \times 10^{-4}$ | $9.4 \times 10^{-4}$ | $6.9 \times 10^{-4}$ | $2.5 \times 10^{-4}$ | $3.4 \times 10^{-4}$ |
| 9 | $5.6 \times 10^{-1}$ | $1.4 \times 10^{-2}$ | $1.8 \times 10^{-3}$ | $4.5 \times 10^{-3}$ | $2.4 \times 10^{-4}$ | $2.8 \times 10^{-5}$ |
| 10 | $3.4 \times 10^{-2}$ | $4.2 \times 10^{-2}$ | $9.9 \times 10^{-4}$ | $3.5 \times 10^{-4}$ | $2.8 \times 10^{-4}$ | $4.1 \times 10^{-5}$ |

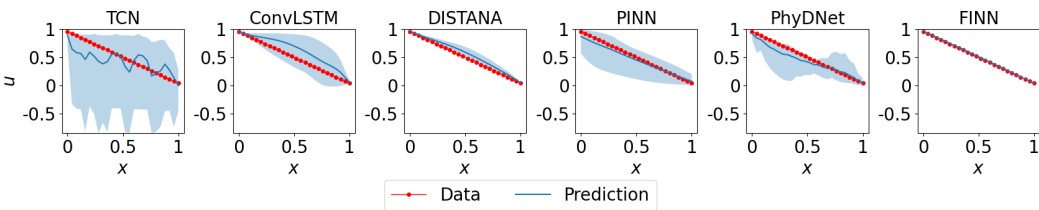

Figure 12: Prediction mean over ten different trained models (with 95% confidence interval) of the dissolved concentration in the diffusion-sorption equation at $t = 10\,000$ for the *in-dis-test* dataset.

Table 7: Closed loop MSE on *out-dis-test* data from ten different training runs for each model for the diffusion-sorption equation.

| Run | TCN | ConvLSTM | DISTANA | PINN | PhyDNet | FINN |
|-----|-----|----------|---------|------|---------|------|
| 1 | $4.7 \times 10^{-2}$ | $2.3 \times 10^{-2}$ | $2.9 \times 10^{-3}$ | - | $1.3 \times 10^{-2}$ | $9.7 \times 10^{-5}$ |
| 2 | $2.1 \times 10^{-1}$ | $4.3 \times 10^{-2}$ | $1.6 \times 10^{-3}$ | - | $1.1 \times 10^{-2}$ | $1.5 \times 10^{-5}$ |
| 3 | $9.2 \times 10^{-2}$ | $5.5 \times 10^{-2}$ | $3.4 \times 10^{-3}$ | - | $1.4 \times 10^{-2}$ | $9.5 \times 10^{-5}$ |
| 4 | $2.0 \times 10^{-1}$ | $1.9 \times 10^{-2}$ | $2.0 \times 10^{-3}$ | - | $1.1 \times 10^{-2}$ | $3.4 \times 10^{-5}$ |
| 5 | $4.4 \times 10^{-2}$ | $1.3 \times 10^{-1}$ | $4.6 \times 10^{-4}$ | - | $2.1 \times 10^{-2}$ | $4.9 \times 10^{-6}$ |
| 6 | $3.9 \times 10^{-4}$ | $1.0 \times 10^{-1}$ | $8.7 \times 10^{-3}$ | - | $4.4 \times 10^{-2}$ | $1.9 \times 10^{-5}$ |
| 7 | $9.3 \times 10^{-4}$ | $9.1 \times 10^{-2}$ | $1.6 \times 10^{-2}$ | - | $1.5 \times 10^{-2}$ | $1.5 \times 10^{-5}$ |
| 8 | $5.0 \times 10^{-3}$ | $2.6 \times 10^{-3}$ | $9.5 \times 10^{-4}$ | - | $1.2 \times 10^{-2}$ | $1.0 \times 10^{-4}$ |
| 9 | $4.6 \times 10^{-1}$ | $3.4 \times 10^{-2}$ | $7.1 \times 10^{-3}$ | - | $1.3 \times 10^{-2}$ | $1.2 \times 10^{-5}$ |
| 10 | $4.1 \times 10^{-2}$ | $8.0 \times 10^{-2}$ | $2.6 \times 10^{-3}$ | - | $1.3 \times 10^{-2}$ | $1.6 \times 10^{-5}$ |

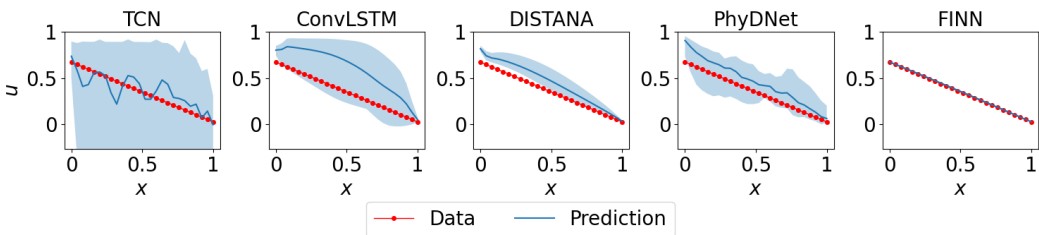

Figure 13: Prediction mean over ten different trained models (with 95% confidence interval) of the dissolved concentration in the diffusion-sorption equation at $t = 10\,000$ for the *out-dis-test* dataset.

defined as a feedforward network with the size of [2, 20, 20, 20, 20, 20, 20, 20, 20, 2]. **PhyDNet** was defined with the PhyCell containing 32 input dimensions, 7 hidden dimensions, 1 hidden layer, and the ConvLSTM containing 32 input dimensions, 32 hidden dimensions, 1 hidden layer. For **FINN**, the modules $\varphi_\mathcal{N}$ and $\varphi_\mathcal{D}$ were used, with $\varphi_\mathcal{D}$ defined as a feedforward network with the size of [1, 10, 20, 10, 1] that takes $u$ as an input and outputs the retardation factor $R(u)$. All models are trained until convergence using the L-BFGS optimizer, except for PhyDNet, which is trained with the Adam optimizer and a learning rate of $1 \times 10^{-3}$ due to stability issues when training with the L-BFGS optimizer.

**Additional results**  Individual errors of the ten different training runs as well as visualizations for the *train* (Table 5), *in-dis-test* (Table 6 and Figure 12), and *out-dis-test* (Table 7 and Figure 13) datasets. Results for the total concentration $u_t$ were omitted due to high similarity to the concentration of the reported contamination solution $u$.

## A.6   DIFFUSION-REACTION

The diffusion-reaction equation is applicable in physical and biological systems, for example in pattern formation (Turing, 1952).

**Data**  In the current paper, we consider the 2D diffusion-reaction for that class of problems:

$$\frac{\partial u_1}{\partial t} = R_1(u_1, u_2) + D_1 \left( \frac{\partial^2 u_1}{\partial x^2} + \frac{\partial^2 u_1}{\partial y^2} \right), \qquad \frac{\partial u_2}{\partial t} = R_2(u_1, u_2) + D_2 \left( \frac{\partial^2 u_2}{\partial x^2} + \frac{\partial^2 u_2}{\partial y^2} \right).$$

$$(11) \qquad\qquad\qquad\qquad (12)$$

Here, $D_1 = 10^{-3}$ and $D_2 = 5 \times 10^{-3}$ are the diffusion coefficient for the activator and inhibitor, respectively. The system of equations is coupled through the reaction terms $R_1(u_1, u_2)$ and $R_2(u_1, u_2)$ which are both dependent on $u_1$ and $u_2$. In this work, the Fitzhugh-Nagumo (Klaasen & Troy, 1984) system was considered to define the reaction function:

$$R_1(u_1, u_2) = u_1 - u_1^3 - k - u_2, \qquad (13) \qquad\qquad R_2(u_1, u_2) = u_1 - u_2, \qquad (14)$$

with $k = 5 \times 10^{-3}$.

The simulation domain for the **train** data is defined with $x = [-1, 1]$, $y = [-1, 1]$, $t = [0, 10]$ and is discretized with $N_x = 49$ and $N_y = 49$ spatial locations, and $N_t = 101$ simulation steps. The initial condition was defined as $u_1(x, 0) = u_2(x, 0) = \sin(\pi(x + 1)/2) \sin(\pi(y + 1)/2)$, and the corresponding boundary condition was defined as $\frac{\partial u_1}{\partial x}(-1, t) = 0$, $\frac{\partial u_1}{\partial x}(1, t) = 0$, $\frac{\partial u_2}{\partial x}(-1, t) = 0$, and $\frac{\partial u_2}{\partial x}(1, t) = 0$.

Table 8: Closed loop MSE on *train* data from ten different training runs for each model for the diffusion-reaction equation.

| Run | TCN | ConvLSTM | DISTANA | PINN | PhyDNet | FINN |
|-----|-----|----------|---------|------|---------|------|
| 1 | $8.7 \times 10^{-3}$ | $1.7 \times 10^{-3}$ | $1.7 \times 10^{-3}$ | $2.7 \times 10^{-4}$ | $8.5 \times 10^{-5}$ | $1.0 \times 10^{-4}$ |
| 2 | $4.4 \times 10^{-3}$ | $1.6 \times 10^{-3}$ | $4.2 \times 10^{-3}$ | $5.6 \times 10^{-4}$ | $7.1 \times 10^{-5}$ | $1.6 \times 10^{-4}$ |
| 3 | $3.2 \times 10^{-2}$ | $1.4 \times 10^{-3}$ | $9.1 \times 10^{-3}$ | $2.1 \times 10^{-4}$ | $5.8 \times 10^{-5}$ | $1.4 \times 10^{-4}$ |
| 4 | $8.7 \times 10^{-3}$ | $7.8 \times 10^{-4}$ | $6.5 \times 10^{-4}$ | $2.8 \times 10^{-4}$ | $6.8 \times 10^{-5}$ | $1.2 \times 10^{-4}$ |
| 5 | $2.5 \times 10^{-2}$ | $1.5 \times 10^{-3}$ | $4.7 \times 10^{-4}$ | $2.8 \times 10^{-4}$ | $7.6 \times 10^{-5}$ | $1.0 \times 10^{-4}$ |
| 6 | $8.0 \times 10^{-3}$ | $4.8 \times 10^{-4}$ | $2.1 \times 10^{-4}$ | $2.5 \times 10^{-4}$ | $6.5 \times 10^{-5}$ | $1.7 \times 10^{-4}$ |
| 7 | $2.4 \times 10^{-2}$ | $3.7 \times 10^{-3}$ | $8.6 \times 10^{-3}$ | $6.1 \times 10^{-5}$ | $8.2 \times 10^{-5}$ | $1.6 \times 10^{-4}$ |
| 8 | $1.1 \times 10^{-2}$ | $1.0 \times 10^{-3}$ | $2.4 \times 10^{-3}$ | $1.0 \times 10^{-4}$ | $7.8 \times 10^{-5}$ | $1.0 \times 10^{-4}$ |
| 9 | $1.3 \times 10^{-2}$ | $2.2 \times 10^{-3}$ | $3.8 \times 10^{-3}$ | $3.2 \times 10^{-5}$ | $8.8 \times 10^{-5}$ | $1.5 \times 10^{-4}$ |
| 10 | $5.6 \times 10^{-3}$ | $7.3 \times 10^{-2}$ | $8.6 \times 10^{-3}$ | $6.2 \times 10^{-4}$ | $7.4 \times 10^{-5}$ | $1.1 \times 10^{-4}$ |

Table 9: Closed loop MSE on *in-dis-test* data from ten different training runs for each model for the diffusion-reaction equation.

| Run | TCN | ConvLSTM | DISTANA | PINN | PhyDNet | FINN |
|---|---|---|---|---|---|---|
| 1 | $1.8 \times 10^{-1}$ | $8.0 \times 10^{-2}$ | $1.6 \times 10^{-1}$ | $7.7 \times 10^{-2}$ | $6.2 \times 10^{-2}$ | $2.9 \times 10^{-3}$ |
| 2 | $5.9 \times 10^{-1}$ | $5.7 \times 10^{-2}$ | $1.9 \times 10^{-1}$ | $2.3 \times 10^{-1}$ | $8.8 \times 10^{-2}$ | $1.6 \times 10^{-3}$ |
| 3 | $6.5 \times 10^{-1}$ | $9.2 \times 10^{-2}$ | $3.0 \times 10^{-1}$ | $3.5 \times 10^{-2}$ | $7.8 \times 10^{-2}$ | $1.7 \times 10^{-3}$ |
| 4 | $3.1 \times 10^{-1}$ | $6.7 \times 10^{-2}$ | $2.0 \times 10^{-1}$ | $2.6 \times 10^{-2}$ | $6.8 \times 10^{-2}$ | $2.9 \times 10^{-3}$ |
| 5 | $7.3 \times 10^{-1}$ | $4.9 \times 10^{-2}$ | $1.9 \times 10^{-1}$ | $1.3 \times 10^{-1}$ | $5.3 \times 10^{-2}$ | $1.7 \times 10^{-3}$ |
| 6 | $1.9 \times 10^{-1}$ | $5.4 \times 10^{-2}$ | $2.8 \times 10^{-2}$ | $8.9 \times 10^{-2}$ | $9.9 \times 10^{-2}$ | $2.0 \times 10^{-3}$ |
| 7 | $6.0 \times 10^{-1}$ | $1.0 \times 10^{-1}$ | $2.2 \times 10^{-1}$ | $5.4 \times 10^{-2}$ | $6.7 \times 10^{-2}$ | $1.5 \times 10^{-3}$ |
| 8 | $7.1 \times 10^{-1}$ | $8.6 \times 10^{-2}$ | $1.9 \times 10^{-1}$ | $1.3 \times 10^{-2}$ | $9.1 \times 10^{-2}$ | $2.3 \times 10^{-3}$ |
| 9 | $5.5 \times 10^{-1}$ | $2.3 \times 10^{-1}$ | $1.6 \times 10^{-1}$ | $2.3 \times 10^{-2}$ | $6.7 \times 10^{-2}$ | $1.8 \times 10^{-3}$ |
| 10 | $2.2 \times 10^{-1}$ | $1.2 \times 10^{-1}$ | $1.7 \times 10^{-1}$ | $2.6 \times 10^{-2}$ | $1.1 \times 10^{-1}$ | $2.1 \times 10^{-3}$ |

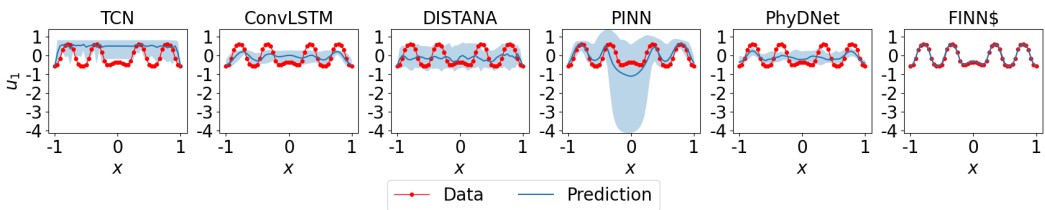

Figure 14: Prediction mean over ten different trained models (with 95% confidence interval) of the activator in the diffusion-reaction equation at $t = 50$ for the *in-dis-test* dataset.

Table 10: Closed loop MSE on *out-dis-test* data from ten different training runs for each model for the diffusion-reaction equation.

| Run | TCN | ConvLSTM | DISTANA | PINN | PhyDNet | FINN |
|---|---|---|---|---|---|---|
| 1 | $2.3 \times 10^{-2}$ | $3.2 \times 10^{-3}$ | $9.4 \times 10^{-3}$ | - | $3.9 \times 10^{-2}$ | $6.5 \times 10^{-3}$ |
| 2 | $1.7 \times 10^{-1}$ | $1.3 \times 10^{-2}$ | $1.3 \times 10^{-2}$ | - | $5.4 \times 10^{-2}$ | $5.5 \times 10^{-3}$ |
| 3 | $2.5 \times 10^{-1}$ | $1.3 \times 10^{-2}$ | $2.1 \times 10^{-2}$ | - | $2.5 \times 10^{-2}$ | $5.9 \times 10^{-3}$ |
| 4 | $7.3 \times 10^{-2}$ | $7.2 \times 10^{-3}$ | $7.9 \times 10^{-3}$ | - | $4.2 \times 10^{-2}$ | $6.5 \times 10^{-3}$ |
| 5 | $2.6 \times 10^{-1}$ | $5.9 \times 10^{-3}$ | $2.6 \times 10^{-3}$ | - | $2.8 \times 10^{-2}$ | $6.1 \times 10^{-3}$ |
| 6 | $6.4 \times 10^{-2}$ | $1.5 \times 10^{-2}$ | $2.1 \times 10^{-3}$ | - | $2.9 \times 10^{-2}$ | $6.0 \times 10^{-3}$ |
| 7 | $2.2 \times 10^{-1}$ | $1.0 \times 10^{-2}$ | $1.9 \times 10^{-2}$ | - | $2.8 \times 10^{-2}$ | $5.7 \times 10^{-3}$ |
| 8 | $2.0 \times 10^{-1}$ | $3.9 \times 10^{-3}$ | $1.2 \times 10^{-2}$ | - | $2.1 \times 10^{-2}$ | $6.1 \times 10^{-3}$ |
| 9 | $2.3 \times 10^{-1}$ | $4.3 \times 10^{-2}$ | $1.3 \times 10^{-2}$ | - | $2.1 \times 10^{-2}$ | $6.0 \times 10^{-3}$ |
| 10 | $3.6 \times 10^{-2}$ | $3.7 \times 10^{-2}$ | $3.3 \times 10^{-2}$ | - | $5.9 \times 10^{-2}$ | $6.2 \times 10^{-3}$ |

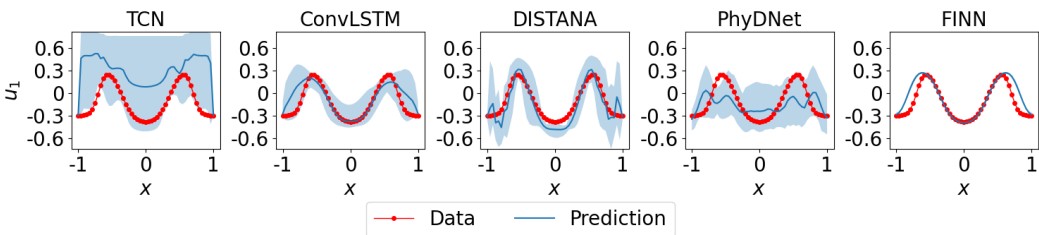

Figure 15: Prediction mean over ten different trained models (with 95% confidence interval) of the activator in the diffusion-reaction equation at $t = 10$ for the *out-dis-test* dataset.

The ***in-dis-test* data** is simulated with with $x = [-1, 1]$, $y = [-1, 1]$ and with the time span of $t = [10, 50]$ and $N_t = 501$. Initial condition is taken from the train data at $t = 10$ and boundary conditions are also identical to the train data.

The simulation domain for the ***out-dis-test* data** was identical with the train data, except for the initial condition that was defined as $u_1(x, 0) = u_2(x, 0) = \sin(\pi(x + 1)/2) \sin(\pi(y + 1)/2) - 0.5$, i.e. subtracting $0.5$ from the original initial condition.

**Model architectures** **TCN** is designed to have two input- and output neurons, and one hidden layer of size 32. **ConvLSTM** has two input- and output neurons and one hidden layer of size 24. The lateral and dynamic input- and output sizes of the **DISTANA** model are set to one and two, respectively, while a hidden layer of size 32 is used. The pure ML models were trained on the first 70 time steps and validated on the remaining 30 time steps of the train data (applying early stopping). Also, to prevent the pure ML models from diverging too much in closed loop, the boundary data are fed into the models as done during teacher forcing. **PINN** is defined as a feedforward network with the size of [3, 20, 20, 20, 20, 20, 20, 20, 20, 2]. **PhyDNet** is defined with the PhyCell containing 32 input dimensions, 49 hidden dimensions, 1 hidden layer, and the ConvLSTM containing 32 input dimensions, 32 hidden dimensions, 1 hidden layer. For **FINN**, the modules $\varphi_\mathcal{N}$, $\varphi_\mathcal{D}$ and $\Phi_\psi$ are used, with $\varphi_\mathcal{D}$ set as two learnable scalars that learn the diffusion coefficients $D_1$ and $D_2$, and $\Phi_\psi$ defined as a feedforward network with the size of [2, 20, 20, 20, 2] that takes $u_1$ and $u_2$ as inputs and outputs the reaction functions $R_1(u_1, u_2)$ and $R_2(u_1, u_2)$. All models are trained until convergence using the L-BFGS optimizer, except for PhyDNet, which is trained with the Adam optimizer and a learning rate of $1 \times 10^{-3}$ due to stability issues when training with the L-BFGS optimizer.

**Additional results** Individual errors of the ten different training runs as well as visualizations for the *train* (Table 8), *in-dis-test* (Table 9 and Figure 14), and *out-dis-test* (Table 10 and Figure 15) datasets. Results for the total inhibitor $u_2$ were omitted due to high similarity to the reported activator $u_1$.

## A.7 ALLEN-CAHN

The Allen-Cahn equation is commonly employed in reaction-diffusion systems, in particular to model phase separation in multi-component alloy systems (Raissi et al., 2019).

**Data** The 1D Allen-Cahn equation is written as

$$\frac{\partial u}{\partial t} = D\frac{\partial^2 u}{\partial x^2} + R(u),$$  (15)

where the main unknown is $u$, the reaction term is denoted as $R(u)$ which is a function of $u$ and the diffusion coefficient is $D = 10^{-4}$. In the current work, the reaction term is defined as:

$$R(u) = 5u - 5u^3,$$  (16)

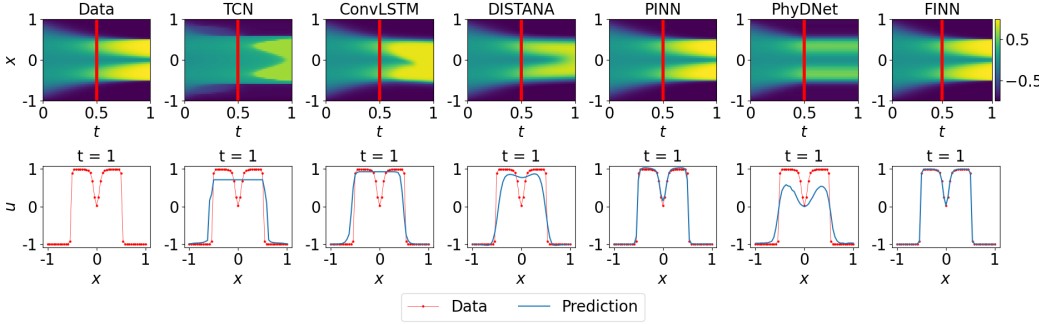

Figure 16: Plots of Allen-Cahn's data and prediction of *in-dis-test* data using different models. The plots in the first row show the solution over $x$ and $t$ (the red lines mark the transition from *train* to *in-dis-test*), the second row visualizes the best model's solution distributed in $x$ at $t = 1$.

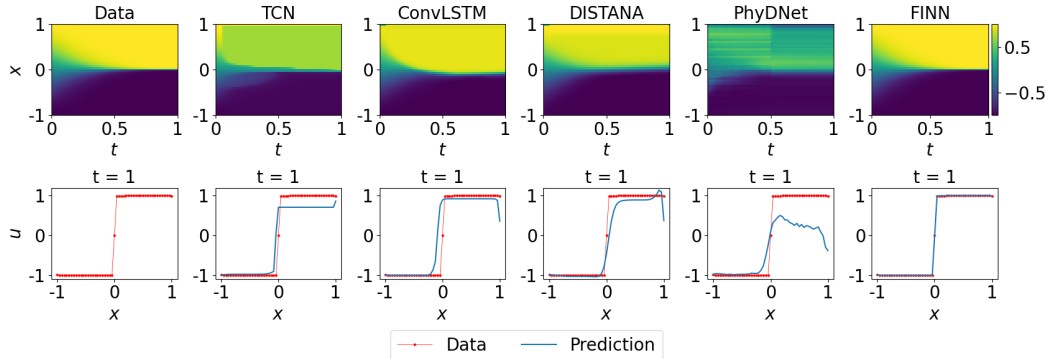

Figure 17: Plots of Allen-Cahn's data and prediction of *out-dis-test* data using different models. The plots in the first row show the solution over $x$ and $t$, the second row visualizes the solution distributed in $x$ at $t = 1$. Right wagga

to reproduce the experiment conducted in the PINN paper (Raissi et al., 2019).

The simulation domain for the ***train* data** is defined with $x = [-1, 1]$, $t = [0, 0.5]$ and is discretized with $N_x = 49$ spatial locations, and $N_t = 201$ simulation steps. The initial condition is defined as $u(x, 0) = x^2 \cos(\pi x)$, and periodic boundary condition is used, i.e. $u(-1, t) = u(1, t)$.

***In-dis-test* data** is simulated with $x = [-1, 1]$ and a time span of $t = [0.5, 1]$ and $N_t = 401$. Initial condition is taken from the train data at $t = 0.5$ and boundary conditions are also similar to the train data.

The simulation domain for the ***out-dis-test* data** is identical with the train data, except for the initial condition that is defined as $u(x, 0) = \sin(\pi x/2)$.

**Model architectures** Both **TCN** and **ConvLSTM** are designed to have one input neuron, one hidden layer of size 32, and one output neuron. The lateral and dynamic input- and output sizes of the **DISTANA** model are set to one and a hidden layer of size 32 is used. The pure ML models were trained on the first 150 time steps and validated on the remaining 50 time steps of the train data (applying early stopping). Also, to prevent the pure ML models from diverging too much in closed loop, the boundary data are fed into the models as done during teacher forcing. **PINN** was defined as a feedforward network with the size of [2, 20, 20, 20, 20, 20, 20, 20, 20, 1] (8 hidden layers, each contains 20 hidden neurons). **PhyDNet** was defined with the PhyCell containing 32 input dimensions, 7 hidden dimensions, 1 hidden layer, and the ConvLSTM containing 32 input dimensions, 32 hidden dimensions, 1 hidden layer. For **FINN**, the modules $\varphi_\mathcal{N}$, $\varphi_\mathcal{D}$, and $\Phi_\psi$ were used, with $\varphi_\mathcal{D}$ defined as a learnable scalar that learns the diffusion coefficient $D$, and $\Phi_\psi$ defined

Table 11: Closed loop MSE on the *train* data from ten different training runs for each model for the Allen-Cahn equation.

| Run | TCN | ConvLSTM | DISTANA | PINN | PhyDNet | FINN |
|-----|-----|----------|---------|------|---------|------|
| 1 | $1.1 \times 10^{-2}$ | $7.1 \times 10^{-2}$ | $2.9 \times 10^{-4}$ | $1.6 \times 10^{-5}$ | $1.5 \times 10^{-4}$ | $1.8 \times 10^{-5}$ |
| 2 | $4.5 \times 10^{-2}$ | $3.0 \times 10^{-2}$ | $4.7 \times 10^{-4}$ | $4.2 \times 10^{-5}$ | $6.4 \times 10^{-5}$ | $8.2 \times 10^{-6}$ |
| 3 | $2.9 \times 10^{-2}$ | $1.2 \times 10^{-2}$ | $8.0 \times 10^{-4}$ | $4.2 \times 10^{-5}$ | $2.5 \times 10^{-4}$ | $5.4 \times 10^{-6}$ |
| 4 | $8.9 \times 10^{-2}$ | $6.5 \times 10^{-3}$ | $4.9 \times 10^{-4}$ | $1.6 \times 10^{-6}$ | $4.8 \times 10^{-5}$ | $4.7 \times 10^{-7}$ |
| 5 | $1.4 \times 10^{-2}$ | $8.0 \times 10^{-2}$ | $2.6 \times 10^{-4}$ | $1.3 \times 10^{-5}$ | $5.1 \times 10^{-5}$ | $2.3 \times 10^{-5}$ |
| 6 | $1.1 \times 10^{-2}$ | $4.6 \times 10^{-3}$ | $5.7 \times 10^{-4}$ | $6.4 \times 10^{-6}$ | $4.7 \times 10^{-5}$ | $2.5 \times 10^{-6}$ |
| 7 | $9.6 \times 10^{-3}$ | $7.1 \times 10^{-3}$ | $4.2 \times 10^{-4}$ | $1.4 \times 10^{-5}$ | $1.7 \times 10^{-4}$ | $1.1 \times 10^{-5}$ |
| 8 | $3.6 \times 10^{-2}$ | $5.3 \times 10^{-4}$ | $8.5 \times 10^{-4}$ | $2.1 \times 10^{-5}$ | $3.7 \times 10^{-5}$ | $2.9 \times 10^{-7}$ |
| 9 | $6.9 \times 10^{-1}$ | $4.0 \times 10^{-1}$ | $7.6 \times 10^{-4}$ | $2.0 \times 10^{-5}$ | $9.8 \times 10^{-5}$ | $3.0 \times 10^{-7}$ |
| 10 | $2.8 \times 10^{-2}$ | $1.8 \times 10^{-4}$ | $1.5 \times 10^{-4}$ | $3.8 \times 10^{-5}$ | $9.8 \times 10^{-5}$ | $2.9 \times 10^{-7}$ |

Table 12: Closed loop MSE on *in-dis-test* data from ten different training runs for each model for the Allen-Cahn equation.

| Run | TCN | ConvLSTM | DISTANA | PINN | PhyDNet | FINN |
|---|---|---|---|---|---|---|
| 1 | $9.9 \times 10^{-2}$ | $1.2 \times 10^{0}$ | $4.8 \times 10^{-2}$ | $7.0 \times 10^{-3}$ | $1.4 \times 10^{-1}$ | $5.6 \times 10^{-5}$ |
| 2 | $2.9 \times 10^{-1}$ | $8.6 \times 10^{-1}$ | $9.1 \times 10^{-2}$ | $1.8 \times 10^{-1}$ | $8.2 \times 10^{-2}$ | $2.4 \times 10^{-5}$ |
| 3 | $2.3 \times 10^{-1}$ | $3.6 \times 10^{-1}$ | $7.9 \times 10^{-2}$ | $4.3 \times 10^{-1}$ | $1.0 \times 10^{-1}$ | $1.6 \times 10^{-5}$ |
| 4 | $4.7 \times 10^{-1}$ | $3.4 \times 10^{-1}$ | $7.4 \times 10^{-2}$ | $4.6 \times 10^{-3}$ | $9.9 \times 10^{-2}$ | $2.1 \times 10^{-6}$ |
| 5 | $1.9 \times 10^{-1}$ | $5.5 \times 10^{-1}$ | $7.3 \times 10^{-2}$ | $2.2 \times 10^{-3}$ | $8.5 \times 10^{-2}$ | $9.3 \times 10^{-5}$ |
| 6 | $1.2 \times 10^{-1}$ | $3.2 \times 10^{-1}$ | $1.0 \times 10^{-1}$ | $2.0 \times 10^{-3}$ | $1.2 \times 10^{-1}$ | $1.1 \times 10^{-5}$ |
| 7 | $1.2 \times 10^{-1}$ | $5.4 \times 10^{-1}$ | $1.3 \times 10^{-1}$ | $5.0 \times 10^{-4}$ | $2.3 \times 10^{-1}$ | $3.4 \times 10^{-5}$ |
| 8 | $3.0 \times 10^{-1}$ | $5.6 \times 10^{-1}$ | $9.5 \times 10^{-2}$ | $1.3 \times 10^{-1}$ | $1.0 \times 10^{-1}$ | $1.3 \times 10^{-6}$ |
| 9 | $1.4 \times 10^{0}$ | $1.3 \times 10^{0}$ | $1.6 \times 10^{-1}$ | $1.2 \times 10^{-2}$ | $1.0 \times 10^{-1}$ | $1.3 \times 10^{-6}$ |
| 10 | $2.3 \times 10^{-1}$ | $5.5 \times 10^{-2}$ | $4.6 \times 10^{-2}$ | $3.3 \times 10^{-2}$ | $9.4 \times 10^{-2}$ | $1.3 \times 10^{-6}$ |

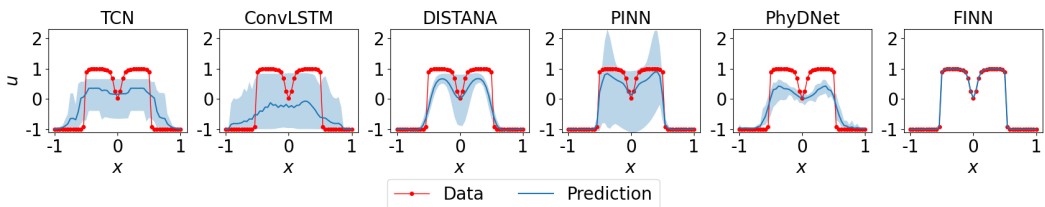

Figure 18: Prediction mean over ten different trained models (with 95% confidence interval) of the Allen-Cahn equation at $t = 1$ for the *in-dis-test* dataset.

Table 13: Closed loop MSE on *out-dis-test* data from ten different training runs for each model for the Allen-Cahn equation.

| Run | TCN | ConvLSTM | DISTANA | PINN | PhyDNet | FINN |
|---|---|---|---|---|---|---|
| 1 | $4.1 \times 10^{-2}$ | $5.3 \times 10^{-2}$ | $1.4 \times 10^{-2}$ | - | $6.2 \times 10^{-1}$ | $7.4 \times 10^{-5}$ |
| 2 | $2.0 \times 10^{-1}$ | $3.8 \times 10^{-1}$ | $3.7 \times 10^{-2}$ | - | $2.4 \times 10^{-1}$ | $2.9 \times 10^{-5}$ |
| 3 | $1.3 \times 10^{-1}$ | $1.5 \times 10^{-1}$ | $6.2 \times 10^{-2}$ | - | $6.2 \times 10^{-1}$ | $1.8 \times 10^{-5}$ |
| 4 | $2.0 \times 10^{-1}$ | $2.9 \times 10^{-1}$ | $1.1 \times 10^{-2}$ | - | $7.3 \times 10^{-1}$ | $3.0 \times 10^{-6}$ |
| 5 | $1.7 \times 10^{-1}$ | $8.3 \times 10^{-2}$ | $4.0 \times 10^{-2}$ | - | $6.8 \times 10^{-1}$ | $1.2 \times 10^{-4}$ |
| 6 | $8.7 \times 10^{-2}$ | $2.9 \times 10^{-1}$ | $3.1 \times 10^{-2}$ | - | $7.8 \times 10^{-1}$ | $1.5 \times 10^{-5}$ |
| 7 | $1.0 \times 10^{-1}$ | $1.5 \times 10^{-1}$ | $5.9 \times 10^{-2}$ | - | $1.0 \times 10^{0}$ | $4.3 \times 10^{-5}$ |
| 8 | $1.5 \times 10^{-1}$ | $5.3 \times 10^{-1}$ | $9.1 \times 10^{-2}$ | - | $5.4 \times 10^{-1}$ | $1.9 \times 10^{-6}$ |
| 9 | $1.2 \times 10^{0}$ | $1.4 \times 10^{0}$ | $1.0 \times 10^{-1}$ | - | $6.2 \times 10^{-1}$ | $1.8 \times 10^{-6}$ |
| 10 | $9.6 \times 10^{-2}$ | $7.5 \times 10^{-2}$ | $1.5 \times 10^{-2}$ | - | $5.9 \times 10^{-1}$ | $1.9 \times 10^{-6}$ |

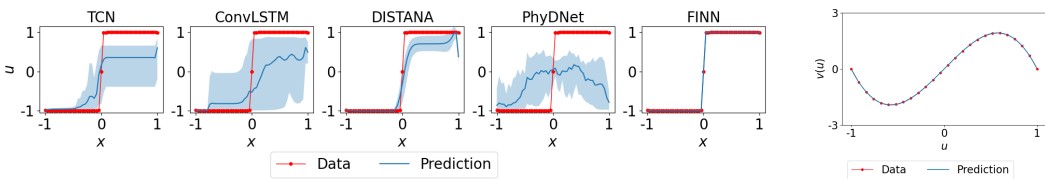

Figure 19: Prediction mean over ten different trained models (with 95% confidence interval) of the Allen-Cahn equation at $t = 1$ for the *out-dis-test* data. The right-most plot shows the accurately learned reaction function from the data samples.

as a feedforward network with the size of [1, 10, 20, 10, 1] that takes $u$ as an input and outputs the reaction function $R(u)$. All models are trained until convergence using the L-BFGS optimizer, except for PhyDNet, which is trained with the Adam optimizer and a learning rate of $1 \times 10^{-3}$ due to stability issues when training with the L-BFGS optimizer.

**Additional results**   Individual errors of the ten different training runs are reported as well as visualizations are generated for the *train* (Table 11), *in-dis-test* (Table 12, Figure 16 and Figure 18), and *out-dis-test* (Table 13, Figure 17 and Figure 19) datasets. Moreover, the right-most plot in Figure 19 shows the dataset's

## A.8   BASELINE ASSESSMENT WITH POLYNOMIAL REGRESSION

In this section, we apply polynomial regression to show that the example problems chosen in this work (i.e. Burger's, diffusion-sorption, diffusion-reaction, and Allen-Cahn) are not easy to solve. First, we use polynomial regression to fit the unknown variable $u = f(x, t)$, similar to PINN. Figure 20 shows the prediction of $u$ for each example, obtained using the fitted polynomial coefficients. For the Burger's equation, the *train* and *in-dis-test* predictions have MSE values of $5.0 \times 10^{-2}$ and $4.1 \times 10^{-2}$, respectively. For the diffusion-sorption equation, the *train* and *in-dis-test* predictions have MSE values of $3.0 \times 10^{-3}$ and $4.1 \times 10^{1}$, respectively. For the diffusion-reaction equation, the *train* and *in-dis-test* predictions have MSE values of $1.7 \times 10^{-2}$ and $3.9 \times 10^{4}$, respectively.

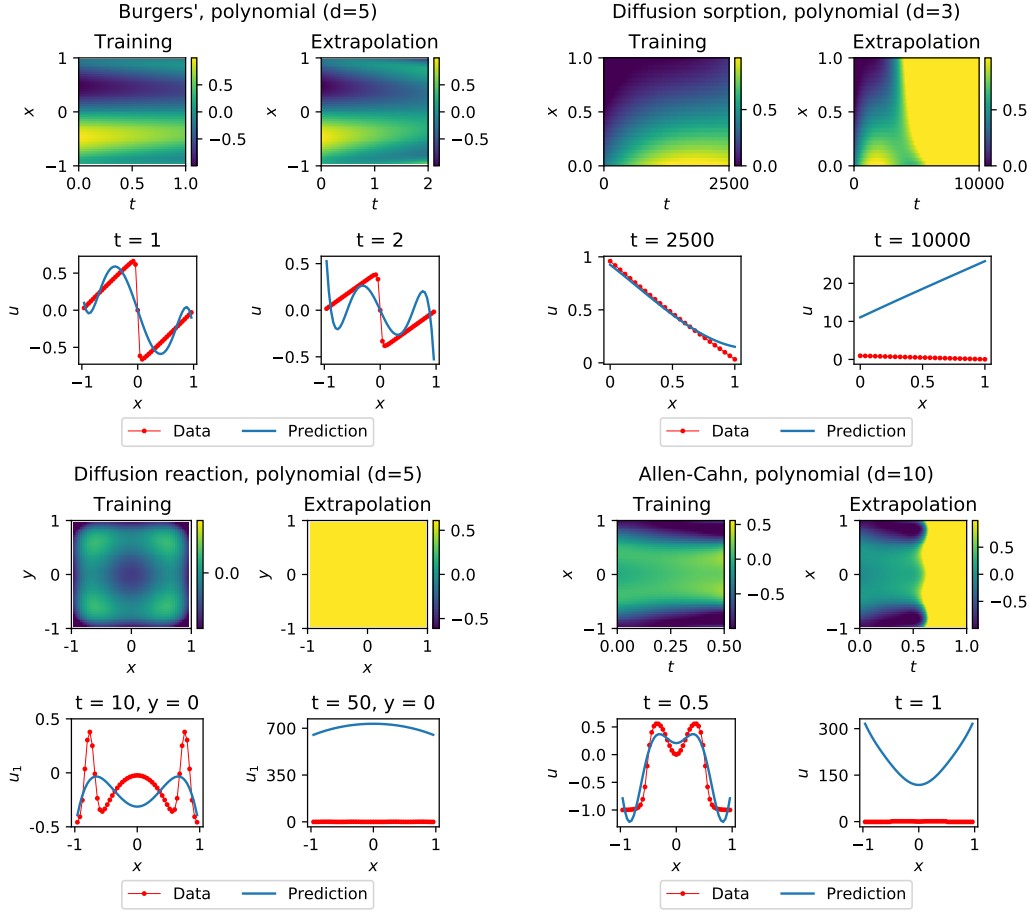

Figure 20: Prediction pairs for *train* and *in-dis-test* (left and right figure columns of the pairs) of the Burger's (left top, polynomial order 5), diffusion-sorption (top right, polynomial order 3), diffusion-reaction (bottom left, polynomial order 5 since higher orders did not converge), and Allen-Cahn (bottom right, polynomial order 10) equations using polynomial regression.

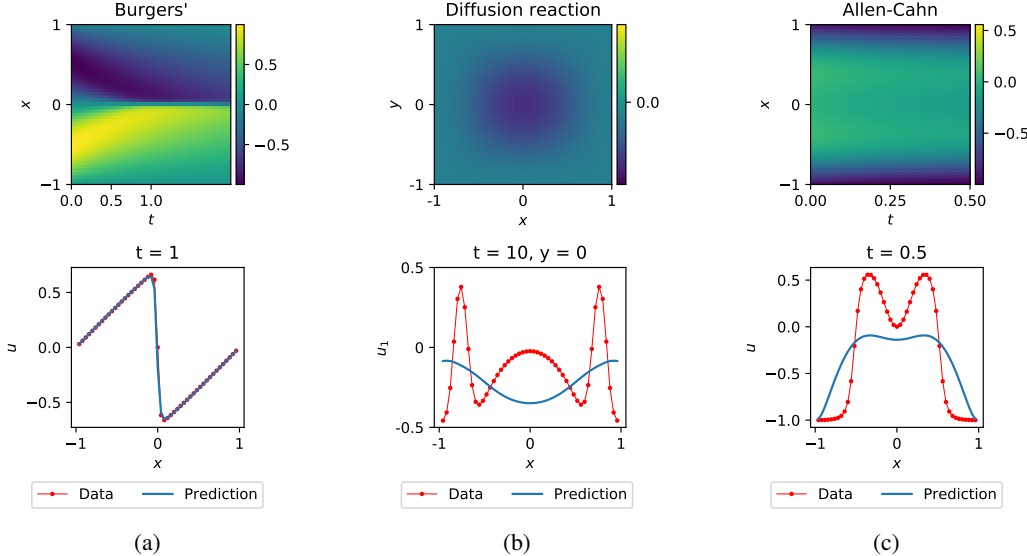

Figure 21: Plots of the *train* prediction in the Burger's (left), diffusion-reaction (center), and Allen-Cahn equations (right) using FINN with polynomial fitting. Due to instability issues, the diffusion-sorption equation could not be solved with the polynomial FINN. The plots in the first row show the solution over $x$ and $t$, and the plots in the second row show the solution distributed in $x$.

For the Allen-Cahn equation, the *train* and *in-dis-test* predictions have MSE values of $8.8 \times 10^{-3}$ and $2.2 \times 10^{3}$, respectively. The simple polynomial fitting fails to obtain accurate predictions of the solution for all example problems. The results also show that the polynomials overfit the data, evidenced by the significant deterioration of performance during extrapolation (prediction of *in-dis-test* data). The diffusion-reaction and the Allen-Cahn equations are particularly the most difficult to fit, because they require higher order polynomials to obtain reasonable accuracy. With the high order, they still fail to even fit the *train* data well.

Next, we also consider using polynomial fitting in lieu of ANNs (namely the modules $\varphi_{\mathcal{A}}$, $\varphi_{\mathcal{D}}$, and $\Phi_{\psi}$) in FINN. With this method, the model successfully obtain accurate prediction of the Burger's equation (Figure 21a). The MSE values are $1.9 \times 10^{-4}$, $1.0 \times 10^{-3}$, and $1.3 \times 10^{-4}$ for *train*, *in-dis-test*, and *out-dis-test* data, respectively. However, the model fails to complete the training for the diffusion-sorption equation due to major instabilities (the polynomials can produce negative output and therefore, negative diffusion coefficient, leading to numerical instability). Moreover, the model also fails to learn the diffusion-reaction (Figure 21b) and the Allen-Cahn (Figure 21c) equations with good accuracy. For the diffusion-reaction equation, the MSE values are $2.5 \times 10^{-2}$, $1.7 \times 10^{-1}$, and $4.8 \times 10^{-2}$ for *train*, *in-dis-test*, and *out-dis-test* data, respectively. For the Allen-Cahn equation, the MSE values are $5.6 \times 10^{-2}$, $2.5 \times 10^{-1}$, and $4.3 \times 10^{-1}$ for *train*, *in-dis-test*, and *out-dis-test* data, respectively. Even though the unknown equations do not seem too complicated, it is still difficult to solve them together with the PDE as a whole. The results show that for these particular problems, ANNs serve better because they allow better control during training, for example with constraints, and they produce more regularized outputs than high order polynomials. However, ANNs are not unique in their selection, but they are more convenient for our implementation.

## A.9  ROBUSTNESS TEST OF FINN

In this section, we test the robustness of FINN when trained using noisy data. All the synthetic data is generated with the same parameters, only added with noise with the distribution $\mathcal{N}(0.0, 0.05)$. For the Burger's equation (Figure 22a and Figure 23a), the average MSE values are $2.5 \times 10^{-3} \pm 4.1 \times 10^{-6}$, $2.4 \times 10^{-3} \pm 6.6 \times 10^{-6}$, and $2.5 \times 10^{-3} \pm 4.0 \times 10^{-6}$ for the *train*, *in-dis-test*, and *out-dis-test* prediction, respectively. For the diffusion-sorption equation (Figure 22b and Figure 23b), the average MSE values are $2.5 \times 10^{-3} \pm 4.5 \times 10^{-6}$, $2.5 \times 10^{-3} \pm 3.7 \times 10^{-6}$, and $2.5 \times 10^{-3} \pm 3.7 \times 10^{-6}$ for the *train*, *in-dis-test*, and *out-dis-test* prediction, respectively. For the diffusion-reaction equation

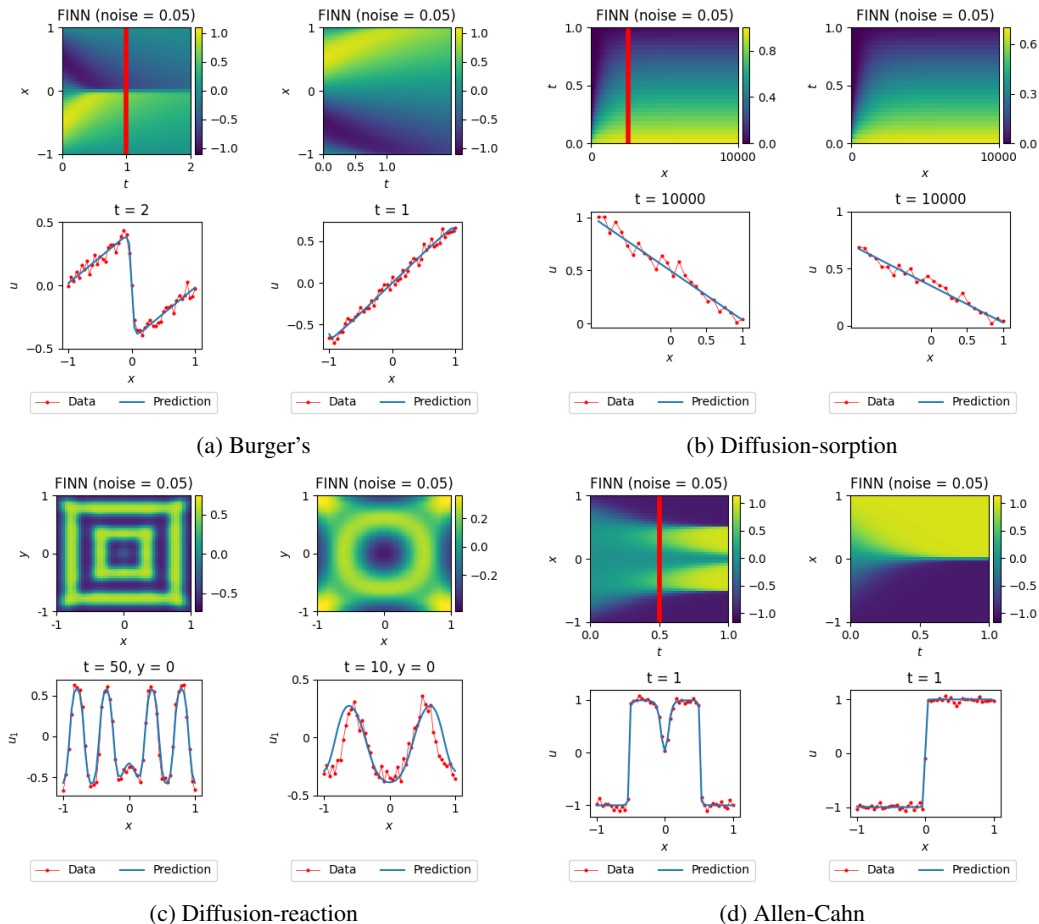

Figure 22: Paired plots of the *in-dis-test* and *out-dis-test* prediction (left and right of the pairs, respectively) in the Burger's (top left), diffusion-sorption (top right), diffusion-reaction (bottom left), and Allen-Cahn (bottom right) equations obtained by training FINN with noisy data. The plots in the first row of the pairs show the solution over $x$ and $t$ (red line marks the transition from *train* to *in-dis-test*), and the plots in the second row of the pairs show the best model's solution distributed in $x$.

(Figure 22c and Figure 23c), the average MSE values are $3.2 \times 10^{-3} \pm 5.3 \times 10^{-4}$, $1.6 \times 10^{-2} \pm 8.8 \times 10^{-3}$, and $8.3 \times 10^{-3} \pm 4.9 \times 10^{-4}$ for the *train*, *in-dis-test*, and *out-dis-test* prediction, respectively. For the Allen-Cahn equation (Figure 22d and Figure 23d), the average MSE values are $2.5 \times 10^{-3} \pm 1.1 \times 10^{-6}$, $2.5 \times 10^{-3} \pm 9.1 \times 10^{-6}$, and $2.5 \times 10^{-3} \pm 6.3 \times 10^{-6}$ for the *train*, *in-dis-test*, and *out-dis-test* prediction, respectively. These results show that even though FINN is trained with noisy data, it is still able to capture the essence of the equation and generalize well to different initial and boundary conditions. Additionally, the prediction is consistent, shown by the low values of the MSE standard deviation, as well as the very narrow confidence interval in the plots.

## A.10 TRAINING PINN WITH FINER SPATIAL RESOLUTION

In order to determine whether the reduced accuracy of PINN in our experiments was caused by a coarse spatial resolution (we only used 49, 26 and $49 \times 49$ spatial positions at Burger, diffusion-sorption, and diffusion-reaction, respectively), another experiment was conducted per target equation where the spatial resolution was increased to $N_x = 999$, $N_x = 251$, and $N_x = 99$, $N_y = 99$, respectively. As reported in Table 14, Figure 24, and Figure 25a, the performance increased slightly but by far did not reach FINN's accuracy. Identical results were achieved in the Burger's and diffusion-sorption equations but are omitted due to high conceptual similarity.

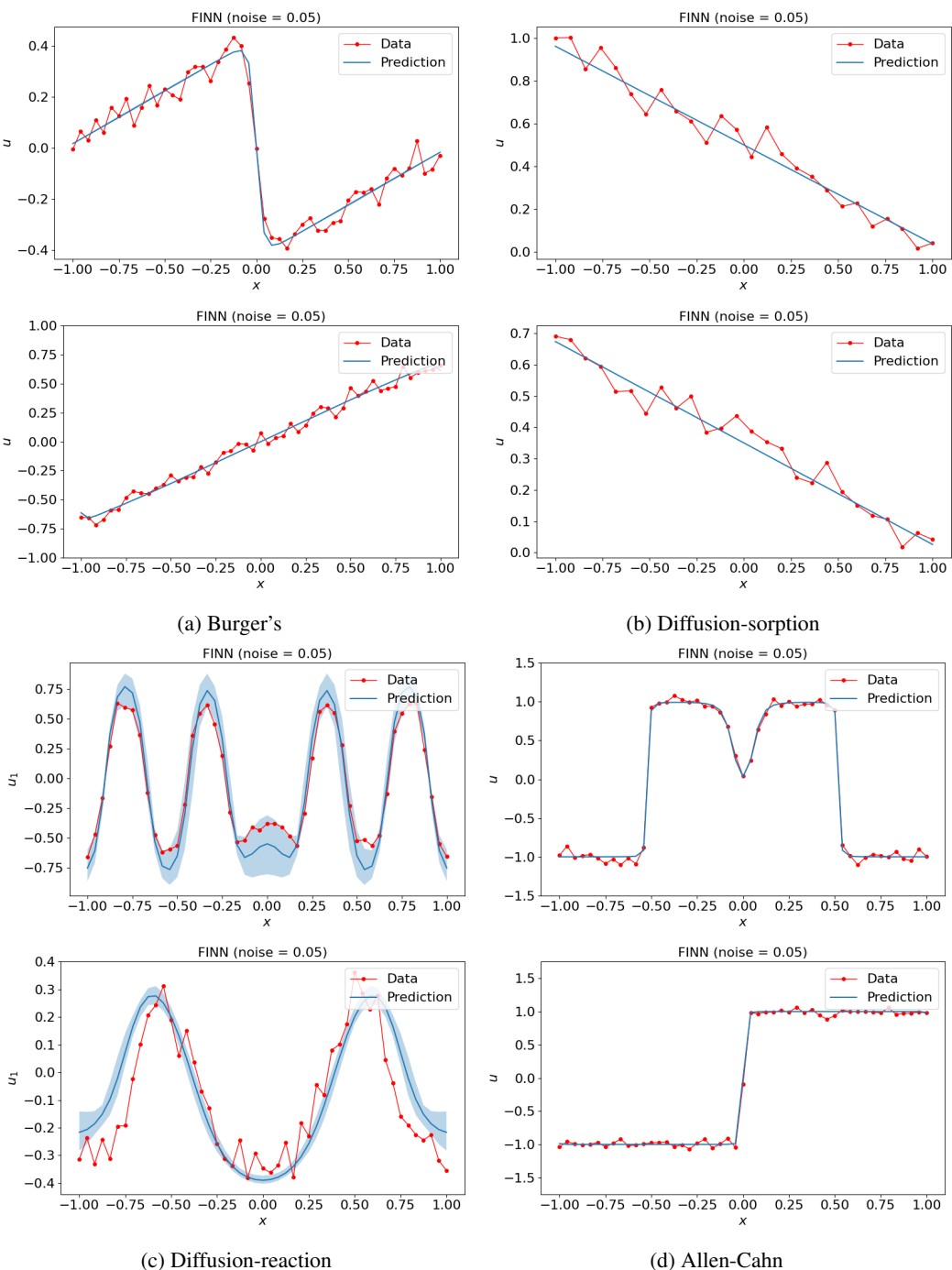

(a) Burger's

(b) Diffusion-sorption

(c) Diffusion-reaction

(d) Allen-Cahn

Figure 23: Prediction mean over ten different trained FINN (with 95% confidence interval) of the Burger's (top left), diffusion-sorption (top right), diffusion-reaction (bottom left), and Allen-Cahn (bottom right) equations obtained by training FINN with noisy data for the *in-dis-test* and *out-dis-test* (top and bottom, accordingly) prediction.

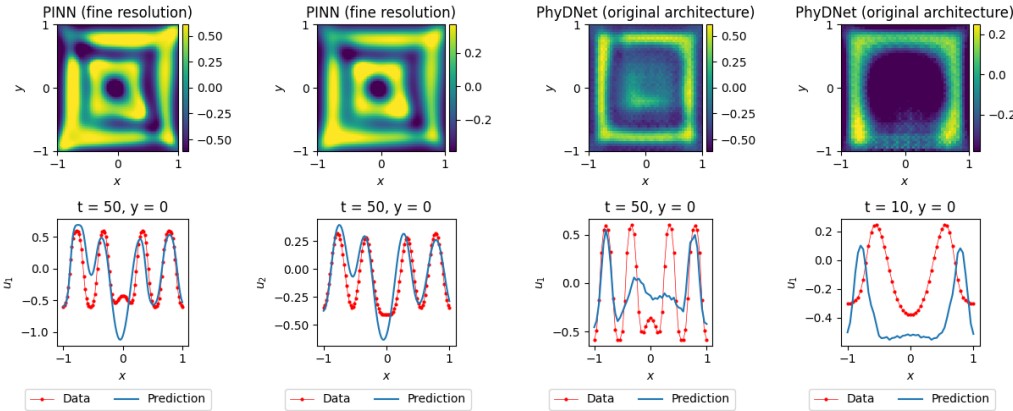

(a) *In-dis-test* prediction (left) and *out-dis-test* (right)   (b) *In-dis-test* prediction (left) and *out-dis-test* (right)

Figure 24: Both left: plots of the diffusion-reaction equation's activator $u$ using PINN trained with finer resolution dataset. Both right: plots of the diffusion-reaction equation's activator $u$ using PhyDNet with the original network size.

Table 14: MSE of PINN trained using data with finer resolution from ten different training runs for the diffusion-reaction equation.

Table 15: MSE of PhyDNet using the original network size from ten different training runs for the diffusion-reaction equation.

| Run | *Train* | *In-dis-test* | *Out-dis-test* | Run | *Train* | *In-dis-test* | *Out-dis-test* |
|---|---|---|---|---|---|---|---|
| 1 | $1.4{\times}10^{-4}$ | $7.8{\times}10^{-2}$ | - | 1 | $9.3{\times}10^{-5}$ | $1.8{\times}10^{-1}$ | $7.6 \times 10^{-2}$ |
| 2 | $1.8{\times}10^{-4}$ | $5.6{\times}10^{-2}$ | - | 2 | $2.9{\times}10^{-5}$ | $6.5{\times}10^{-2}$ | $3.5 \times 10^{-2}$ |
| 3 | $8.6{\times}10^{-5}$ | $1.5{\times}10^{-2}$ | - | 3 | $2.2{\times}10^{-5}$ | $6.7{\times}10^{-2}$ | $4.0 \times 10^{-2}$ |
| 4 | $1.0{\times}10^{-3}$ | $1.5{\times}10^{-1}$ | - | 4 | $4.6{\times}10^{-5}$ | $7.2{\times}10^{-2}$ | $3.9 \times 10^{-2}$ |
| 5 | $5.2{\times}10^{-5}$ | $2.7{\times}10^{-2}$ | - | 5 | $3.5{\times}10^{-5}$ | $6.5{\times}10^{-2}$ | $8.8 \times 10^{-2}$ |
| 6 | $2.9{\times}10^{-4}$ | $8.1{\times}10^{-1}$ | - | 6 | $5.7{\times}10^{-5}$ | $1.2{\times}10^{-1}$ | $1.0 \times 10^{-1}$ |
| 7 | $1.3{\times}10^{-4}$ | $3.4{\times}10^{-2}$ | - | 7 | $3.5{\times}10^{-5}$ | $5.7{\times}10^{-2}$ | $9.0 \times 10^{-2}$ |
| 8 | $8.9{\times}10^{-5}$ | $2.9{\times}10^{-2}$ | - | 8 | $3.5{\times}10^{-5}$ | $5.9{\times}10^{-2}$ | $2.6 \times 10^{-2}$ |
| 9 | $3.9{\times}10^{-5}$ | $1.3{\times}10^{-2}$ | - | 9 | $2.3{\times}10^{-5}$ | $6.3{\times}10^{-2}$ | $4.1 \times 10^{-2}$ |
| 10 | $3.3{\times}10^{-5}$ | $1.7{\times}10^{-2}$ | - | 10 | $3.5{\times}10^{-5}$ | $6.1{\times}10^{-2}$ | $8.1 \times 10^{-2}$ |

## A.11   PHYDNET WITH ORIGINAL AMOUNT OF PARAMETERS

To verify whether our reduction of parameters and the removal of the encoder and decoder layers caused PhyDNet to perform worse, we repeated the experiments for the three equations of interest using the original PhyDNet architecture as proposed in Guen & Thome (2020). However, our results indicate no significant changes in performance, as reported in Table 15, Figure 24b, and Figure 25b. Again, results for the inhibitor $u_2$ as well as for the Burger's and diffusion-reaction equations were omitted due to high conceptual similarity.

## A.12   SOIL PARAMETERS AND SIMULATION DOMAINS FOR THE EXPERIMENTAL DATASET

Identical to Praditia et al. (2021), the soil parameters and simulation (and experimental) domain used in the real-world diffusion-sorption experiment are summarized in Table 16 for core samples #1, #2, and #2B.

For all experiments, the core samples are subjected to a constant contaminant concentration at the top $u_s$, which can be treated as a Dirichlet boundary condition numerically. Notice that, for core sample #2, we set $u_s$ to be slightly higher to compensate for the fact that there might be fractures at the top of core sample #2, so that the contaminant can break through the core sample faster.

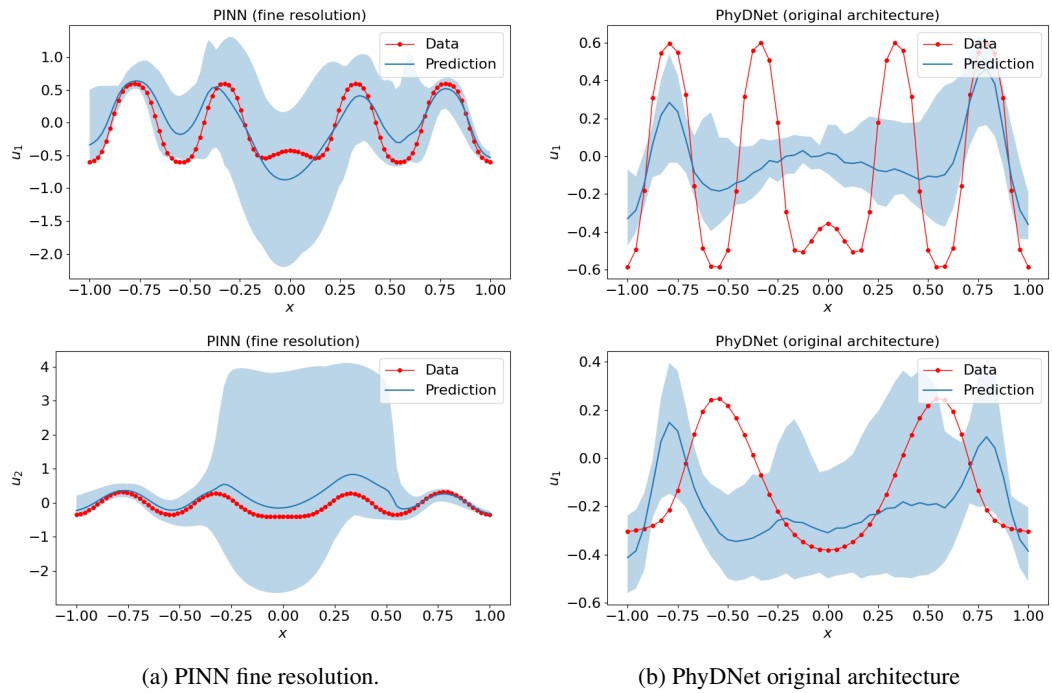

(a) PINN fine resolution.  (b) PhyDNet original architecture

Figure 25: Prediction mean (with 95% confidence interval) of the activator (top) and inhibitor (bottom) in the diffusion-reaction equation at $t = 50$ compared with the *in-dis-test* data.

For core samples #1 and #2, $Q$ is the flow rate of clean water at the bottom reservoir that determines the Cauchy boundary condition at the bottom of the core samples. For core sample #2B, note that the sample length is significantly longer than the other samples, and by the end of the experiment, no contaminant has broken through the core sample. Therefore, we assume the bottom boundary condition to be a no-flow Neumann boundary condition; see (Praditia et al., 2021) for details.

Table 16: Soil and experimental parameters of core samples #1, #2, and #2B. $D$ is the diffusion coefficient, $\phi$ is the porosity, $\rho_s$ is the bulk density, $L$ and $r$ are the length and radius of the sample, $t_{end}$ is the simulation time, $Q$ is the flow rate in the bottom reservoir and $u_s$ is the total concentration of trichloroethylene in the sample.

| Soil parameters | | | |
|---|---|---|---|
| Parameter | Unit | Core #1 | Core #2 | Core #2B |
| $D$ | m$^2$/day | $2.00 \times 10^{-5}$ | $2.00 \times 10^{-5}$ | $2.78 \times 10^{-5}$ |
| $\phi$ | - | 0.288 | 0.288 | 0.288 |
| $\rho_s$ | kg/m$^3$ | 1957 | 1957 | 1957 |
| Simulation domain | | | |
| Parameter | Unit | Core #1 | Core #2 | Core #2B |
| $L$ | m | 0.0254 | 0.02604 | 0.105 |
| $r$ | m | 0.02375 | 0.02375 | N/A |
| $t_{end}$ | days | 38.81 | 39.82 | 48.88 |
| $Q$ | m$^3$/day | $1.01 \times 10^{-4}$ | $1.04 \times 10^{-4}$ | N/A |
| $u_s$ | kg/m$^3$ | 1.4 | 1.6 | 1.4 |

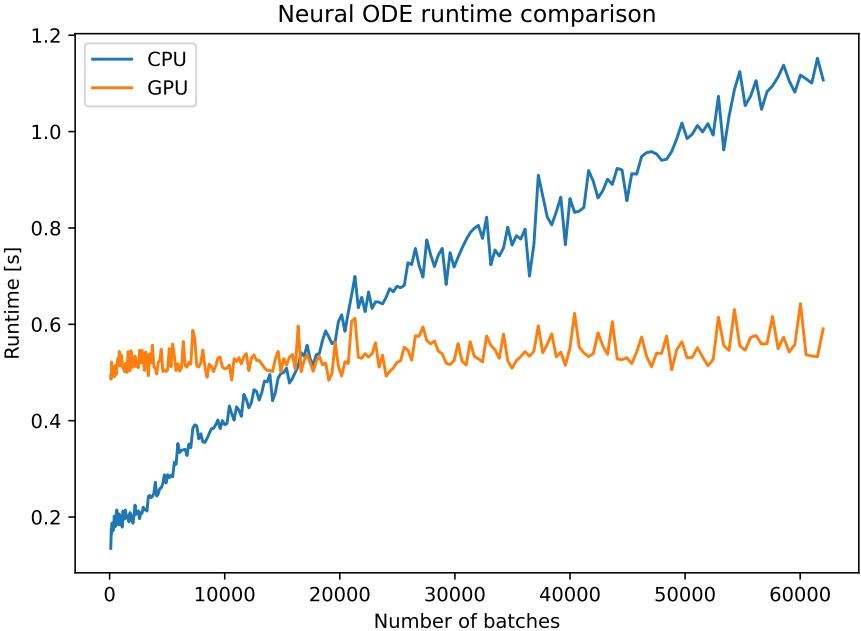

Figure 26: Runtime comparison of Neural ODE with a single hidden layer consisting of 50 hidden nodes run on CPU and GPU for 1 000 time steps. The benefit of using GPU starts to be seen when larger batch sizes (> 20 000) are used.

## A.13 ON TIME STEP ADAPTIVITY AND RUNTIME

We compare the runtime for each model, run on a CPU with i9-9900K core, a clock speed of 3.60 GHz, and 32 GB RAM. Additionally, we also perform the comparison of GPU runtime on a GTX 1060 (i.e. with 6GB VRAM). The results are summarized in Table 17. Note that the purpose of this comparison is not an optimized benchmark, but only to show that the runtime of FINN is comparable with the other model, especially when run in CPU. When run on GPU, however, FINN runs slightly slower. This is caused by the fact that FINN's implementation is not yet optimized for GPU. More importantly, the Neural ODE package we use benefits only from a larger batch size. As shown in Figure 26, GPU is faster for batch size larger than 20 000, whereas the maximum size that we use in the example is 2 401. With smaller batch size, CPU usage is faster.

In general, only the PINN model has a benefit when computed on the GPU, since the function is only called once on all batches. This is different for all other models that have to unroll a prediction of the sequence into the future recurrently (except for TCN which is a convolution approach that is faster on GPU). Accordingly, The overhead of copying the tensors to GPU outweighs the GPU's parallelism benefit, compared to directly processing the sequence on the CPU iteratively. On the two-dimensional diffusion-reaction benchmark, the GPU's speed-up comes into play, since in here, the simulation domain is discretized into $49 \times 49 = 2401$ volumes (compared to 49 for the Burger's and Allen-Cahn, and 26 for the diffusion-sorption equations). Note that we have observed significantly varying runtimes on different GPUs (i.e. up to three seconds for FINN on Burger's on an RTX 3090), which might be caused by lack of support of certain packages for a particular hardware, but further investigation is required.

Additionally, we want to emphasize that the higher runtime of FINN on GPU is not caused by the time step adaptivity. In fact, employing the adaptive time stepping strategy is cheaper than choosing all time steps to be small enough (to guarantee numerical stability). As we learn a PDE, we have no exact knowledge to derive a dedicated time integration scheme, but the adaptive Runge-Kutta method is one of the best generic choices. As our PDE and its characteristics change during training, time step adaptivity is a real asset, because for example the Courant–Friedrichs–Lewy (CFL)

Table 17: Comparison of runtime (in seconds) of single forward passes between different deep learning (above dashed line) and physics-aware neural network (below dashed line) methods on the different equations.

| Eqn. | Model | CPU | GPU (GTX 1060) |
|---|---|---|---|
| Burger | TCN | 0.423 | 0.130 |
| | ConvLSTM | 0.052 | 0.079 |
| | DISTANA | 0.059 | 0.098 |
| | PINN | 0.036 | 0.007 |
| | PhyDNet | 0.107 | 0.192 |
| | FINN | 0.066 | 0.161 |
| Diffusion-sorption | TCN | 1.228 | 0.393 |
| | ConvLSTM | 0.119 | 0.194 |
| | DISTANA | 0.145 | 0.223 |
| | PINN | 0.073 | 0.010 |
| | PhyDNet | 0.263 | 0.475 |
| | FINN | 0.676 | 1.638 |
| Diffusion-reaction | TCN | 14.15 | 0.800 |
| | ConvLSTM | 0.230 | 0.052 |
| | DISTANA | 0.190 | 0.051 |
| | PINN | 1.647 | 0.787 |
| | PhyDNet | 0.159 | 0.113 |
| | FINN | 0.342 | 0.330 |
| Allen-Cahn | TCN | 0.442 | 0.128 |
| | ConvLSTM | 0.050 | 0.082 |
| | DISTANA | 0.060 | 0.097 |
| | PINN | 0.035 | 0.007 |
| | PhyDNet | 0.108 | 0.191 |
| | FINN | 0.028 | 0.071 |

condition (Courant et al., 1967) would consistently change throughout the training. Therefore, the time step adaptivity is not a bottleneck, but rather a solution for a more efficient computation.

Furthermore, in relation to FINN's limitation (see subsection A.2) topics like numerical stabilization schemes for larger time steps, adaptive spatial grid, optimized implementation in an High Performance Computing (HPC) setting, parallelization, etc. are still very interesting for future works.

