# OpenReview forum: "Composing Partial Differential Equations with Physics-Aware Neural Networks"
_ICLR.cc/2022/Conference — ICLR 2022 Submitted_

### Official Review · Reviewer_TVaa · 2021-10-28

**Correctness:** 3
**Technical Novelty And Significance:** 3
**Empirical Novelty And Significance:** 3
**Recommendation:** 6
**Confidence:** 2

**Main Review:**

Strong points:
1.	The authors propose very detailed experimental results, with both synthetic and real data
2.	The background is provided in detail as well

Weak points:
1.	The authors claim in section 3 that convolution layers can only accommodate single types of boundary conditions. However, this is not correct since convolution layers at present can also handle cases I.e. periodic boundary conditions, this sentence may need rewriting
2.	In figure 1, phi_N is shown to be both first and second-order derivatives, which may create confusion for the reader.
3.	The training process is not completely clear, the authors describe the forward process, one would assume backpropagation of error for updating the neural network, but this is not explicit in the text. A figure demonstrating the whole training process, including the feedback loop might be helpful
4.	At the end of section 3, the authors claim to use NODE in place of Euler for the reason of numerical stability, however, NODE also used Euler and does not mention anything about adaptive time-stepping, more details on this part would help to clear things up



**Summary Of The Paper:**

The authors propose to model advection-diffusion partial differential equations as a composition of multiple neural networks. According to the authors, this leads to better generalization over different initial and boundary conditions for the physics system, and also the ability to learn different factors of the process as modeled by the terms in the PDE. Extensive experimental results are presented to support the author’s claims that their framework performs better than state-of-the-art.

**Summary Of The Review:**

The paper has very thorough experimental results and good descriptions of the methods. However, the claims made in the paper may need a second look/ need some rewriting.

---

> ### Author Response · Authors · 2021-11-15
> **Answers to reviewer TVaa**
>
> We thank the reviewer for the constructive feedback and questions which we answer in the following and are looking forward to answering more questions that may arise.
>
> ### About the boundary condition statements with convolutions
> This is indeed correct, we have rewritten the statement in the manuscript. Thank you for pointing it out. The point that we wanted to make originally is that FINN is able to handle and generalize various types of boundary conditions, even a more complicated one that requires derivative values like Neumann and Cauchy.
>
> ### About the confusion about $\varphi_\mathcal{N}$
> We understand the confusion. The module $\varphi_\mathcal{N}$ is indeed used to calculate the spatial derivative, both first and second-order. The difference lies in the application of the $\mathcal{R}$ module for the first-order derivative. We updated Figure 1 for clarification and emphasize this in the text further and provide more details in the general answer to all reviewers above.
>
> ### About lacking information concerning the training process
> We understand the concern. First, the error is calculated for all available $u(x,t)$. The training process then uses backpropagation through time of this error for the parameter updating. Thanks to the fully differentiable NODE solver, the backpropagation can be performed through the solver directly by definition, but an adjoint method can also be used for the backpropagation in case the operations inside the solver is not differentiable. We have added red arrows to Figure 1 indicating the backward flow of gradients and the feedback loop and also added a statement of how backpropagation through time is used to update the weights at the end of section 3. Indeed, this makes the procedure much more illustrative and graspable; we appreciate this suggestion.
>
> ### About adaptive time-stepping
> Adaptive computation is one of the main features highlighted in the original NODE paper by Chen, et al., stating that "Modern ODE solvers provide guarantees about the growth of approximation error, monitor the level of error, and adapt their evaluation strategy on the fly to achieve the requested level of accuracy." Additionally, the default ODE solver used in the package torchdiffeq (https://github.com/rtqichen/torchdiffeq), which we apply, is the Runge-Kutta method with adaptive step size. We have added the remark to the adaptive Runge-Kutta method to the end of section 3.

---

### Official Review · Reviewer_rFkv · 2021-10-29

**Correctness:** 4
**Technical Novelty And Significance:** 3
**Empirical Novelty And Significance:** 2
**Recommendation:** 3
**Confidence:** 3

**Main Review:**

**Strong points:**

The paper provides significant improvement in performance in comparison to other models in a few orders of magnitude using 3-5 times less parameters than related models.

The results also suggest that modelling the spacial derivatives du / dx with the neural network directly results in better generalisation than modelling u(x,t and taking the derivative with respect to x (similar to PINN and other similar works). This is an interesting and counter-intuitive conclusion and worth of further exploration.

**Weak points:**

The paper requires more explanation on the structure of the model. What is the input to phi_N, phi_A  and phi_D? How is R computed (it is not part of the equation in figure 1)? In Figure 1, why does phi_N signify both the first and the second spacial derivatives of u?

If the neural networks phi take only the values u, how are u(x,t) computed at the first time step?

Based on figure 1, the right-hand side of the equation is a combination of four neural network outputs. Do you think it is equivalent to replacing phi_D phi_N - phi_A phi_N with a single neutral network taking all u_{i-1}, u_i, u_{I+1} (which essentially turns it into a Neural ODE model)?

It would be helpful to emphasise the differences between FINN, PINN and PhyDNet and other models in the methods section. Some of this explanation is provided in section 4.1, but it is worth emphasising and explaining more how the models differ, what exactly is modelled by a neural network and the motivation why FINN performs better than other models.


**Questions:**

Page 7: “PINN requires complete knowledge of the modelled system in form of the equation”. In this case, I am puzzled by a worse performance of PINN in figures 6 and 7 compared to FINN, if PINN closely follows the PDE that generated the data. Can you provide some insight why FINN performs better than PINN in this case?

Similarly, can you clarify why PINN requires more parameters than FINN (table 1)? Is it due to the fact that PINN also models the function u(x, t) or because of the bigger network?



**Summary Of The Paper:**

The paper introduces Finite Volume neural network for modelling fluid dynamics inspired by the finite volume method. The paper models the velocity and the spacial derivatives as the neural networks. The work demonstrates more precise fluid simulation than the related physics-inspired models.

**Summary Of The Review:**

The results in comparison to other models are compelling, particularly the model comparison on figures 6 and 7. However, the methods section of the paper needs more clarification and justification of the modelling choices in comparison to other papers. As the paper requires a significant re-write, I suggest a reject.

---

> ### Author Response · Authors · 2021-11-15
> **Answers to reviewer rFkv**
>
> We thank the reviewer for the constructive feedback and questions which we answer in the following and are looking forward to answering more questions that may arise.
>
> ### About clarity of the modules and their inputs
> For calculating $u$ at cell $i$ in time step $t+1$, both $\varphi_\mathcal{A}$ and $\varphi_\mathcal{D}$ take $u$ at cell $i$ and time step $t$ as input. The module $\varphi_\mathcal{N}$ takes $u$ at cell $i$ and time step $t$ as well as $u$ at the neighboring cell $i-1$ and $i+1$ (also at time step $t$) as input.
> $R$ is a module that post-processes the output of $\varphi_\mathcal{A}$ to calculate the advective flux, and is calculated based on equation 3. Because the $R$ module is applied for the first-order spatial derivative only (as specified by the conditional statement in equation 3), $\varphi_\mathcal{N}$ signifies the first-order spatial derivative in the advective flux calculation. Without the $\mathcal{R}$ module, $\varphi_\mathcal{N}$ signifies the second-order spatial derivative relating to the diffusive flux (the integration result is written in equation 4). The inputs to all modules and the information flow are illustrated in Figure 1 and mentioned explicitly in the text, and we have added the module $R$ to the equation in Figure 1 to further clarify this; thank you for the remark.
> More details can be found in the new section A.3 of the appendix. Also, as indicated in the general answer above, we admit the confusion caused by the $\varphi_\mathcal{N}$ module which seemed to learn both the first and the second spatial derivatives and have modified Figure 1 and the accompanying equation accordingly.
>
> ### About computing at the first time step
> At the first time step, we provide an initial condition, that is $u(x,t)$ at time step $t=0$, so that calculation of $u$ at time step $t = 1$ uses this known initial condition. The procedure is analogous to solving PDEs using a numerical solver. Effectively, starting at this initial condition, FINN is applied entirely in closed loop to unfold a prediction into the future (similar to a recurrent neural network). In contrast to other methods, no preceding phase of teacher forcing is required. This information can now be found at the very end of section 3.
>
> ### About modeling with Neural ODE
> The statement is true. In the end, we obtain a system of coupled ODEs. While we can directly model this with Neural ODE without additional inductive bias (only taking all $u_i$, $u_{i-1}$, and $u_{i+1}$ as inputs), the result would not be as good because the network has too much flexibility. This is also discussed in the work of Yin, et al. (2021), where the purely Neural ODE result is outperformed by other models with better inductive bias. We have added this to the end of the new section A.1. of the appendix detailing more differences to related work.
>
> ### About differences to related work
> Thank you for the remark. We added the clarification in the manuscript and, in particular, added a detailed comparison to other methods to the appendix (i.e. section A.1).
>
> ### About PINN being worse than FINN
> This is a very good point. To clarify, the results shown in Figure 6 are the extrapolated prediction. PINN produces predictions with similar accuracy as FINN during training. However, when we try to extrapolate the prediction with temporal domain $t > t_{train}$, PINN's performance starts to deteriorate, meaning that it generalizes worse than FINN.
> Additionally, we have repeated the experiment with three-times the parameters from other experiments (FINN only has twice as many parameters compared to other experiments) and for 1500 instead of 500 epochs -- we observe slight improvement. Still, with less parameters (900 vs 11\,k) and only 100 epochs of training, FINN reaches better results. We see the reason for this in the apparent difficulty of learning the overall equation rather than its constituents with a neural network, which relates to the benefit of implementing reasonable inductive bias. When inspecting the deviations between the ten trained models, we also observe less robustness in PINN in comparison to FINN, as outlined in Figures 10, 12, 14 and 18.
>
> ### About PINN needing more parameters
> As suggested by our re-evaluation from above, we can clearly say that PINN depends on more parameters and longer training in order to reach FINN's performance and agree with your intuition. Since PINN has to directly learn the entire equation along with the correct derivatives, the problem appears to be significantly harder compared to learning the equation's constituents, particularly in the diffusion-reaction problem which consists of two nonlinearly coupled PDEs that have to be modeled by a single PINN. The PINN setup that we use throughout our experiments is the exact same as used in the original paper by Raissi, et al. In the paper, the authors conducted training with a different number of layers and neurons, and we took the best setting.

---

> > ### Comment · Reviewer_rFkv · 2021-11-18
> > **The reply from the reviewer rFkv**
> >
> > Thank you for the clarifications provided about the model (particularly about phi_N), the differences between PINN and FINN and the comment about the Neural ODE baseline. After reading the author's replies, I have another question.
> >
> > It is still unclear to me how the authors convert the PDE into an ODE to be able to run the ODE solver. The authors say that the ODE is a function of t, u_{i-1}, u_i, u_{i+1}, where i is the index of the cells along the spacial dimension(s). If I understand correctly, we have to solve an ODE for every spacial cell indexed by i. Since the cells are dependent on each other, I assume that the authors would have to concatenate all the spacial cells u_i's into one vector to pass it to the ODE Solver. According to section A.10, the authors used 49x49 grid for some  experiments, which would in ~2401-dimensional ODE. This can be problematic with the adaptive-step methods, because the adaptive step will be chosen to be very small, likely resulting in a massive runtime of the method.
> >
> > Can the authors clarify how the input to the ODE solver is constructed across the spacial indices i?
> >
> > Do you observe any limitations of converting a PDE into an ODE, for example, increase in a runtime compared to classic PDE  solvers and other models, like PINN? If so, is there any way to potentially overcome these limitations?

---

> > > ### Author Response · Authors · 2021-11-19
> > > **Second response to reviewer rFkv**
> > >
> > > Thank you for your time and for acknowledging our clarifications, we hope that they are clear now. Please find below our answers to your further comments:
> > >
> > > ### On converting PDE to ODE
> > > Any Eulerian PDE solver (to be more specific, the Finite Volume solver) discretizes the simulation domain spatially to convert the PDE into a system of coupled (time-derivative) ODEs (i.e. the time derivative $\partial u/\partial t$ is calculated separately at each cell $i$), and then uses a time integration scheme to solve the coupled system in parallel. We translate this directly to FINN. FINN is employed recurrently for every spatial cell $i = 1, ..., N_x$. To be more specific, for every cell $i$, FINN takes the input of $u_{i-1}$, $u_i$, and $u_{i+1}$, and then produces the time derivative $\partial u_i/\partial t$. This can also be seen as putting $N_x$ number of "batches" to FINN as inputs (although here, each batch is coupled to its neighbors) and producing $N_x$ batches of time derivatives $\partial u/\partial t$. These time derivatives, at all cells $i$ are then concatenated as an input vector to the Neural ODE. Accordingly, the Neural ODE then takes care of the time integration to obtain $u$ at the following time step (Figure 1). In other words, the Flux Kernels in FINN process the spatial coupling between cells (to calculate the spatial derivative), and the Neural ODE realizes the time integration.
> > > The number of grid points $N_x$ does not influence the determination of the adaptive time step size. Instead, the cell size $\Delta x$ is the main limiting factor, according to some numerical conditions such as the CFL condition (Courant, et. al., 1967). This condition enforces that the time step size should be less than the time for the quantity $u$ to travel to the neighboring cells (i.e. small $\Delta x$ requires small time steps). However, FINN also generalizes well when trained to data with coarse spatial resolution (with larger $\Delta x$ so that the time step size is not restrictive), and it does not need high resolution data, unlike PINN (Appendix A.10). Additionally, adaptive time stepping is cheaper than choosing all time steps to be small enough (to guarantee numerical stability). As we learn a PDE, we have no exact knowledge to derive a dedicated time integration scheme, but the adaptive Runge-Kutta method is one of the best generic choices. As our PDE and its characteristics change during training, time step adaptivity is a real asset, because for example the CFL condition would consistently change throughout the training. Therefore, the time step adaptivity is not a bottleneck, but rather a solution for a more efficient computation. We have added this clarification in the Appendix A.13.
> > >
> > > ### On runtime limitations
> > > We did not observe any limitations of converting a PDE into a system of ODEs by comparing the runtime for each model on a CPU with i9-9900K core, a clock speed of 3.60 GHz, and 32 GB RAM. We also perform the comparison of GPU runtime on a GTX 1060 (i.e. with 6GB VRAM). To clarify, this comparison is only as a demonstration, because the implementation might not be optimized for each model. As shown in the table, FINN's runtime is comparable to other models when run in CPU, showing that solving PDE as a system of coupled ODEs is not a cause for slow computation. When run on GPU, however, FINN computation decreases more compared to other models. We want to emphasize that this higher runtime on GPU is not caused by the adaptive time stepping strategy employed by the ODE solver. Instead, this is caused by the fact that 1.) FINN's implementation is not yet optimized for GPU, and 2.) the Neural ODE package we use benefits only from a larger batch size (larger than 20,000), whereas the maximum size that we use in the example is $49\times 49 = 2,401$ (the 2D diffusion-reaction equation). With smaller batch size, CPU usage is faster (see Appendix A.13 for the plot).
> > > As further improvement, topics like numerical stabilization schemes for larger time steps, adaptive spatial grid, optimized implementation in an HPC setting, parallelization, etc. are still very interesting for future works. We have also added this runtime comparison in the Appendix A.13.
> > >
> > > Eqn. | Model | CPU | GPU (GTX 1060)
> > > --- | --- | --- | ---
> > > Burger's | TCN | 0.423 | 0.130
> > > | ConvLSTM | 0.052 | 0.079
> > > | DISTANA | 0.059 | 0.098
> > > | PINN | 0.036 | 0.007
> > > | PhyDNet | 0.107 |0.192
> > > | FINN | 0.066 | 0.161
> > > Diffusion-sorption | TCN | 1.228 | 0.393
> > > | ConvLSTM | 0.119 | 0.194
> > > | DISTANA | 0.145 | 0.223
> > > | PINN | 0.073 | 0.010
> > > | PhyDNet | 0.263 | 0.475
> > > | FINN | 0.676 | 1.638
> > > Diffusion-reaction | TCN | 14.15 | 0.800
> > > | ConvLSTM | 0.230 | 0.052
> > > | DISTANA | 0.190 | 0.051
> > > | PINN | 1.647 | 0.787
> > > | PhyDNet | 0.159 | 0.113
> > > | FINN | 0.342 | 0.330
> > > Allen-Cahn | TCN | 0.442 | 0.128
> > > | ConvLSTM | 0.050 | 0.082
> > > | DISTANA | 0.060 | 0.097
> > > | PINN | 0.035 | 0.007
> > > | PhyDNet | 0.108 | 0.191
> > > | FINN | 0.028 | 0.071
> > >
> > > Please let us know if you have any further questions.

---

### Official Review · Reviewer_LWqu · 2021-11-02

**Correctness:** 4
**Technical Novelty And Significance:** 2
**Empirical Novelty And Significance:** Not applicable
**Recommendation:** 6
**Confidence:** 3

**Main Review:**

Strengths

This method can not only deal with smooth solutions of general PDE but also can deal with weak solutions of hyperbolic problems, which is a good point; It is also compared with many recent SOTA methods and is significantly ahead of recently published methods

Weaknesses

The method limits the form of the equation, which will greatly increase the training complexity when it is extended to higher-dimensional problems

The problems studied by this method are relatively simple and are the fitting of some linear problems or low-order nonlinear problems. I am wondering if polynomial fitting will get better results.

Some test examples can be supplemented. Such as equations containing an exponential function, the fitted coefficient containing singularity, or can this method be used for fluid equations, etc.

The robustness test of this method is absent. Can this method be used for noisy data?

The discussion of method limitation is absent.


**Summary Of The Paper:**

This paper presents a compositional physics-aware neural network (FINN) for learning spatiotemporal advection-diffusion processes. It claims that the FINN outperforms pure machine learning and other state-of-the-art physics-aware models in all cases—often even by multiple orders of magnitude. However, the design of the network depends too much on the form of the equation, which leads to a very narrow application of the method.

**Summary Of The Review:**

The demonstration of this method is pretty good, but there are too many restrictions on the problem. If the form of the equation is fixed, a simple polynomial fitting may achieve a better fitting effect than the neural networks.

---

> ### Author Response · Authors · 2021-11-15
> **Answers to reviewer LWqu**
>
> We thank the reviewer for the constructive feedback and questions which we answer in the following and are looking forward to answering more questions that may arise.
>
> ### About the limitations of equations
> While we agree that the model requires a prior knowledge of the modeled equation to some extent, we do not see this as a limiting factor. Instead, we argue that our method exploits only the very fundamental principles that are well-known and traditionally used to build a partial differential equation, thus aiding the learning process of the model (via inductive learning biases). Similar approaches have been used in the works of Raissi, et al. (2019), Yin, et al. (2021) and Kochkov, et al. (2021). Even though our method is limited to second-order partial derivatives, there are many physical problems that can be modeled in this way, including the diffusion of fluid, biological pattern formation, population growth, groundwater flow, heat conduction, etc.
> Furthermore, we agree that our current method suffers an increase of training complexity when extending to higher-dimensional data, but since this is an issue of not only most physics-aware neural networks but also for traditional neural networks, we do not perceive this as a unique drawback of our method. Apart from the scaling argument in the conclusion, we have now pointed this out more clearly in an explicit section A.2 in the appendix about FINN's limitation. However, in contrast to all other methods, the number of parameters for FINN hardly increases when switching from 1D to 2D domains, still yielding the most accurate results.
>
> ### About polynomial fitting
> Thank you for addressing this. In fact we do not consider all of the evaluated problems as simple. We followed your suggestion and tried to solve them with polynomial regression. While the results of these basic methods are reasonable on the Burger's equation when they are used to fit the unknown variable $u = f(x,t)$, they perform significantly worse on all other problems (see section A.8 in the appendix, i.e. Figure 20). We argue that this is mainly due to the lack of physical inductive bias. Additionally, these basic methods will also fail to generalize to different boundary and initial conditions because the function $u = f(x,t)$ will change with different boundary and initial conditions.
> However, if the intention is to use the polynomial regression for learning the unknown relationship/equation, then we argue that the learning paradigm used in our method (ANN) could of course be replaced by other regression models. In our method, we choose ANNs as the specific learning paradigm, because we just need one that accepts regularizations and constraints and is as shape-free as possible. ANNs serve well for this purpose, but are not unique in their selection. An analysis where we replaced the ANN modules in FINN by polynomial fitting can be found in section A.8 of the appendix (i.e. Figure 21) and justifies the choice of ANNs over polynomials.
>
> ### About supplementing test examples
> The retardation factor function used to generate the diffusion-sorption data is indeed an exponential function (equation 10 in the Appendix), with $u^{(n_f-1)}$. Additionally, since $n_f < 1$, then $R(u) \rightarrow \infty$ for $u \rightarrow 0$, which is in fact a singularity. In the Burger's example, the solution domain contains a jump at $x = 0$, which is also not easy to approximate. The same holds for the solution to the Allen-Cahn equation, which contains multiple jumps. Regarding the applicability, we show that our method can be used for various cases, especially for modeling fluid dynamics in the diffusion-sorption example. We show this using both synthetic datasets and real-world data (Section 4.2). Moreover, FINN is also able to model pattern formation in the diffusion-reaction example. We now also point out the challenge of modeling our chosen equations, since simple approaches such as polynomial fitting did not succeed. Also, as reported in the general answer above, we have conducted another series of experiments on the Allen-Cahn equation with a cubic exponential term, confirming our former results (section A.7 in the appendix).
>
> ### About the robustness of our method
> Thank you for the suggestion. This method is applicable also for noisy data. This is shown in Section 4.2, where we train FINN with a real experimental dataset which is noisy, and where we are able to generalize it to the other samples. Additionally, we performed another training of FINN with noise of 5\% added to the synthetic data. These additional results are presented in the section A.9 and Figure 23 of the appendix and we show that FINN's prediction is still relatively accurate and, more importantly, physically plausible and consistent
>
> ### About missing limitations
> This is a very good point, thank you for the suggestion. We have added a detailed section A.2 in the appendix discussing FINN's limitations.

---

> > ### Comment · Reviewer_LWqu · 2021-11-18
> > **update**
> >
> > I'm glad you added a series of ablation tests to make the model more robust. Thank you.

---

> > > ### Author Response · Authors · 2021-11-18
> > > **Further suggestions or concerns?**
> > >
> > > Thank you for your time and for acknowledging our ablation tests. Please note that, apart from the ablations, we have additionally added another test example to demonstrate our model's applicability to a further equation (i.e. the Allen-Cahn equation), we have added cases with more noise (which you have addressed with your statement about making our model more robust), and we have complemented the manuscript with a section about limitations of FINN.
> > >
> > > With our extensions and answers, we hope to have addressed all points you have mentioned or criticized. Do you have any more comments or suggestions on how we may improve our work further?

---

### Official Review · Reviewer_j4js · 2021-11-03

**Correctness:** 3
**Technical Novelty And Significance:** 3
**Empirical Novelty And Significance:** 3
**Recommendation:** 6
**Confidence:** 4

**Main Review:**

This is a very concrete paper with solid experiments.
To better assess this work, I would like to ask several questions:

1. I find it misleading to call this model "compositional neural network", which usually refers to composing neural network as f \cdot g = f(g). If I understand correctly, it is not what the paper does.

2. If I understand correctly, the FINN model computes the derivative using the finite-difference method. It can be understood as a learn FDM/FVM stencil. I wonder how it compared to other learned FDM methods such as Machine learning–accelerated computational fluid dynamics by Brenner et. al.?

3. If there is a spectrum of ML models vs conventional numerical solvers, I think this FINN method lies closer to the end of numerical solvers (which is nothing bad). Solver-like methods usually require fewer parameters and generalize better. Therefore, I think it could be valuable if the authors can add some numerical experiments comparing against the numerical solver. Are there any advantages (speed, accuracy, etc) to using the learn FVM vs the original FVM/FDM?

4. Standard advection-diffusion equations are relatively easy to solve. I wonder if the idea proposed in this paper can be generalized and transferred to other PDEs?


**Summary Of The Paper:**

In this work, the authors propose the finite volume neural network to solve advection-diffusion partial differential equations. The authors define a flux kernel as the sum of sub-kernel f_i on each element. And the authors define specific modules \phi_D, \phi_A, \phi_N according to the form of the advection-diffusion equation. The model requires very few parameters to achieve the state of art results.

**Summary Of The Review:**

I think the paper is above the threshold. If the authors can provide evidence that the proposed method outperforms (or has some relative advantage) compared to standard FVM solvers, I will be happy to raise the score.

---

> ### Author Response · Authors · 2021-11-15
> **Answers to reviewer j4js**
>
> We thank the reviewer for the constructive feedback and questions which we answer in the following and are looking forward to answering more questions that may arise.
>
> ### About the term "compositional"
> Thank you for the comment. We think this might be a misunderstanding. A serially composition of ANN modules may be the simplest form of a compositional ANN. Generally speaking we intended to refer to compositional ANNs as ANNs that are processing input in a distributed, modularized, compositional manner. Stemming from the linguistics side originally, the term has been used rather widely in the ANN literature and related literature in cognitive science. We thus added a few references to our introduction clarifying the terminology and hope that the reviewer agrees, that we can keep it as is.
>
> ### About comparison with the work of Brenner et al.
> Yes, this is correct. FINN learns the Finite Volume (and in case of regular grids, this becomes the Finite Difference) stencil. However, this is not the main feature of FINN. The works of the group around Brenner and Hoyer employ a similar idea, adding inductive bias into the model, but it uses this bias mainly to improve interpolation from coarse to finer grid resolution. Thus, Brenner et al. aim at accelerating simulation, whereas our work focuses on discovering unknown relationships/laws (or re-discovering laws in the case of the synthetic examples), such as the advective velocity in the Burgers' example, the retardation factor function in the diffusion-sorption example, and the reaction function in the diffusion-reaction example.
> Additionally, the works by Brenner et al. employ a convolutional structure, which is only applicable to Dirichlet or periodic boundary conditions, and it suffers from a slight instability during training when the training data trajectory is unrolled for a longer period. In contrast to being convolutional, FINN employs the flux kernel, calculated at all control volume surfaces, which enables the implementation for discovery of various boundary conditions and of unknown constituents such as retardation factors through the precise control of the information flow. Additionally, FINN employs the Neural ODE method as the time integrator to reduce numerical instability during training with long time series. FINN is also able to generalize well when trained with a relatively sparse dataset (not only coarse resolution, but only slices in time or space), reducing the computational burden and data demand further. We now emphasize these points explicitly in the manuscript by including comparisons to these works in the new section A.1 to the appendix in order to clarify the contribution of our manuscript.
>
> ### About the difference to FVM/FDM solvers
> It is difficult to compare FINN vs FVM when using synthetic datasets, because FINN is trained with data generated with the FVM solver. However, we can show that FINN becomes superior compared to the standard FVM solver (also called the "physical model" in the manuscript), when we apply it to a real-world dataset (which we can only approximate with PDEs and assumptions), as presented in Section 4.2. Figure 9 shows the prediction of FINN and the physical model. In the main text, we furthermore show that FINN improves performance by at least halfing the MSE in various core samples.
>
> ### About the application to other PDEs
> In our work, we also apply FINN to a diffusion-sorption problem as well as a diffusion-reaction problem. Nevertheless, we have conducted another series of experiments on the Allen-Cahn equation, as detailed in the general answer above. However, it is still limited to PDEs with second-order partial derivatives. But, even though our method is limited only to second-order partial derivative, there are many physical problems that can be modelled using PDEs up to second-order partial derivative, such as the diffusion of fluid, biological pattern formation, population growth, groundwater flow, heat conduction, etc. We have now clarified this in the problem formulation of section 3 of our manuscript.

---

> > ### Comment · Reviewer_j4js · 2021-11-25
> > **I am not very certain about the contribution.**
> >
> > My major concern was how the proposed FVM method compares with the original FVM solver. The author partially answered my question. There seems to be some but vague improvement, but I am not sure how significant it is. I will hold my score but lower my confidence level. I am willing to follow other reviewers' opinions.

---

> > > ### Author Response · Authors · 2021-11-26
> > > **Second answer to reviewer j4js**
> > >
> > > Thank you for your response. To re-emphasize, the FVM solver is used to generate the datasets for the synthetic examples. Therefore, the FVM solver itself will always be 100% accurate because it is considered as the "ground truth" (i.e. reference data), while the comparison between FINN and the FVM solver in a real-world scenario leads to the MSE presented in section 4.2 of the manuscript. The comparison will only make sense when we have an application to a non-synthetic dataset.
> > >
> > > Moreover, in the PDE, there are many unknowns such as the advective velocity, diffusion coefficient, retardation factor, reaction function. To match the data, modelers have to make certain assumptions when building the FVM solver, i.e. the choice of unknown correlation like the sorption isotherm to calculate the retardation factor (linear, Freundlich, or Langmuir) and the choice of reaction function (Fitzhugh-Nagumo, Gierer-Meinhardt, Gray-Scott, etc.). Additionally, the modelers also have to calibrate the unknown empirical parameters that define the chosen correlation. With FINN, we replace these unknowns with modules that serve as universal approximators, which reduces the necessity for additional assumptions (that might be wrong) to the bare minimum. Examples are shown in Figure 8, where FINN successfully learns the advective velocity, retardation factor, and reaction functions with relatively high accuracy for the synthetic case. In the real-world example, FINN learns the retardation factor when trained only with data coming from a single core sample, and generalizes very well to other core samples even with different boundary condition (Figure 9), as well as outperforming the FVM solver. More precisely, in the system of the targeted application domain (where model assumptions for solvers are not available or inaccurate), FINN outperforms the solver which, in this situation, is struggling with its discrete selection from the available set of possible model assumptions that are all inappropriate for the poorly understood system.
> > >
> > > We hope that we have addressed your concerns about the main difference between FINN and FVM solvers, which boils down to higher accuracy in modeling real-world problems and the elimination of the subjective experimentation required to find a suitable FVM setup (given an understanding of the process is available at all). Please let us know if you have further concerns, we appreciate your feedback.

---

### Author Response · Authors · 2021-11-15
**General answer to all reviewers**

## About the concerns that the equations are too simple

We appreciate the suggestion to benchmark our method on another challenging equation and chose Allen-Cahn with its cubic reaction term (see below) as the fourth benchmark. Note that we indeed interpret the three already benchmarked equations as challenging and not straight forward to solve (highly nonlinear at $x=0$ in Burger's, exponent and singularity in diffusion-sorption, 2D diffusion-reaction scenario with cubic term in equation 13 and with two non-linearly coupled PDEs). In fact, the challenging nature of these equations could be confirmed by additional analysis suggested by reviewer LWqu, where we have tried to solve all three existing equations by means of polynomial fitting and only succeeded for Burger's on the \textit{in-dis-test} data. The Allen-Cahn equation is defined as
\begin{equation}
    \frac{\partial u}{\partial t} = D\frac{\partial^2 u}{\partial x^2} + R(u),
\end{equation}
where the main unknown is $u$, the reaction term is denoted as $R(u)$ which is a function of $u$ and the diffusion coefficient is $D = 10^{-4}$. Results on learning this equation are reported in Appendix A.7 of the revised manuscript and confirm our results, where FINN again outperforms all other methods. For comparability, the reaction term is defined similar to Raissi et al. (2019) as:
\begin{equation}
    R(u) = 5u - 5u^3.
\end{equation}

## About whether $\varphi_{\mathcal{N}}$ approximates first and second spatial derivative

The short answer is no, but we admit that our original figure encouraged this interpretation. Semantically, the $\varphi_\mathcal{N}$ module learns the numerical stencil, that is the geometrical arrangement of a group of neighboring cells to approximate the derivative numerically, effectively learning the first spatial derivative $\frac{\partial u}{\partial x}$ from both $[u_{i-1}, u_i]$ and $[u_{i}, u_{i+1}]$, which are the inputs to the $\varphi_ {\mathcal{N}i-}$ and $\varphi_{\mathcal{N}i+}$ module, respectively.

The lateral information flowing from $u_{i-1}$ and $u_{i+1}$ toward $u_i$ is controlled by the $\varphi_\mathcal{A}$ (advective flux, i.e. bulk motion of many particles/atoms that can either move to the left or to the right) and the $\varphi_\mathcal{D}$ (diffusive flux, i.e. drive of particles/atoms to equilibrium from regions of high to low concentration) modules. Since the advective flux can only move either to the left or to the right, it will be considered only in the left flux kernel ($f_{i-}$) or in the right flux kernel ($f_{i+}$), and not both at the same time. The case-sensitive ReLU module $\mathcal{R}$ (Equation 3) decides on this, by setting the advective flux in the irrelevant flux kernel to zero (effectively depending on the sign of the output of $\varphi_\mathcal{A}$). Thus, the advective flux is only considered from either $u_{i-1}$ or $u_{i+1}$ to $u_i$, which amounts to the first order spatial derivative.

The diffusive flux, on the other hand, can propagate from both sides towards the control volume of interest $u_i$ and, hence, the second order spatial derivative, accounting for the difference between $u_{i-1}$ and $u_{i+1}$, has to be applied. In our method, this is realized through the combination of the $\varphi_\mathcal{N}$ and $\varphi_\mathcal{D}$ modules, calculating the diffusive fluxes $\delta_- = \varphi_\mathcal{N}(u_{i-1}, u_i)\varphi_\mathcal{D}(u_i)$ and $\delta_+ = \varphi_\mathcal{N}(u_i, u_{i+1})\varphi_\mathcal{D}(u_i)$ inside of the respective left and right flux kernel. The combination of these two deltas in the state kernel ensures the consideration of the diffusive fluxes from left (including $u_{i-1}$) and from right (including $u_{i+1}$), resulting in the ability to account for the second order spatial derivative.

Technically, the first, i.e. $[-1, 1]$, and second, i.e. $[1, -2, 1]$, order spatial differentiation schemes are common definitions and a derivation can be found, for example, in Fornberg's ''Generation of Finite Difference Formulas on Arbitrarily Spaced Grids" (1988).
However, a quick derivation of the Laplace scheme $[1, -2, 1]$ can be formulated as follows. Define the second order spatial derivative as
\begin{equation}
    \frac{\partial^2 u}{\partial x^2} \approx \frac{(\partial u/\partial x)|^{i-} - (\partial u/\partial x)|^{i+}}{\Delta x}
\end{equation}

with $(\partial u/\partial x)|^{i-} \approx (u_{i-1} - u_i)/\Delta x$ and $(\partial u/\partial x)|^{i+} \approx (u_i - u_{i+1})/\Delta x$. Then
\begin{equation}
    \frac{\partial^2 u}{\partial x^2} \approx \frac{(u_{i-1} - u_i) - (u_i - u_{i+1})}{\Delta x^2} = \frac{u_{i-1} - 2u_i + u_{i+1}}{\Delta x^2}
\end{equation}
hence the $[+1, -2, +1]$ as coefficients. Thank you for addressing this point, we hope the reformulations of section 3 along with the new section A.3 we have added to the Appendix increase the comprehensibility.

---

### Author Response · Authors · 2021-11-15
**Updated manuscript and supplementary material**

We have updated the manuscript with the suggestions made by the reviewers. Code and material to reproduce our additional experiments can be found in the updated supplementary material.

Below we provide a list of major changes we made to the manuscript:
* We clarified the applicability of the PDE with up to second order spatial derivative in the Problem Formulation of Section 3.
* We clarified the inputs for all modules $\varphi_\mathcal{N}$, $\varphi_\mathcal{A}$, and $\varphi_\mathcal{D}$ in the Flux Kernel subsection of Section 3.
* We updated Figure 1 for better clarity, including the equation in Figure 1 to remove confusion about $\varphi_\mathcal{N}$, and added red arrows to identify the flow of gradient during backpropagation.
* We added the Allen-Cahn equation as a supplement to the experiment, also to show the wider range of applicability of our method. We also add the information in the abstract, and the results are shown in Appendix A.7.
* We added a detailed explanation of distinction to and comparison with related works in Appendix A.1.
* We added a detailed discussion of FINN's limitation in Appendix A.2.
* We added a detailed explanation of the numerical stencil learned by the module $\varphi_\mathcal{N}$ in Appendix A.3.
* We added a baseline assessment, comparing the result of polynomial fitting to solve the example equations in Appendix A.8.
* We added a robustness test of FINN to show that it still generalizes well even when trained with noisy data in Appendix A.9.
* We added a runtime comparison of all models and a corresponding discussion in Appendix A.13.

---

### Decision · Program_Chairs · 2022-01-20

**Decision:**

Reject

**Comment:**

Thank you for your submission to ICLR.  There is some disagreement about this paper, and several of the reviews are of relatively low confidence.  While I appreciate the effort that the authors have put into addressing the concerns of the reviewers, after going through the paper and the responses myself, I'm ultimately coming down on the side of the less positive reviews.  My reasoning, honestly, is that I think the authors are vastly overestimating the knowledge that the ICLR audience will have about numerical methods for PDE solutions.  Reading through the paper, I honestly have very little idea about how the actual numerical techniques are carried out, and it's unclear to me precisely where this method falls in between a traditional numerical solver an actual neural network.  Reading through the reviews, even the more positive ones, I don't think I'm alone in this perception (and the authors will hopefully believe me that these reviewers _are_ indeed emblematic of the subgroup of ICLR that is most experience with differential equations).  I really feel like either a substantial rewrite of the paper is needed, to make clear the full extent of the numerical methods being applied; or alternatively, the work may really be better suited for a numerical methods venue.

---

> ### Public Comment · ~Shuhao_Cao1 · 2022-01-29
> **Summary from a computational PDE amateur researcher for the community of ML**
>
> >  Reading through the paper, I honestly have very little idea about how the actual numerical techniques are carried out, and it's unclear to me precisely where this method falls in between a traditional numerical solver an actual neural network.
>
> Popular ML approaches approximate the solution function $u$, this paper uses NN to approximate the flux, i.e., $u'$, or $-c(u_x, u_y)$ in 2D. The reasons are straightforward (but not obvious to researchers having not worked on PDE-related problems). Flux ($u'$) are physical because they are from the conservation laws (mass, momentum), while the potentials/densities ($u$) are less physical. Even in the arena of traditional methods for fluid problems (reservoir simulation), Finite Volume is used a lot more than Finite Element due to its nice conservation property. Poincare inequality determines that the approximation error $u$ is bounded by that in $u'$ (with some boundary condition requirement), so if one approximates $u'$ better, normally $u$ can be accordingly approximated better.
>
> However, the elephant in the room is, the authors did not mention that Finite Volume Method itself can achieve much higher accuracy than FINN under the same total FLOPs budget. I am not talking about the number of params in an NN (less params, less local mins, harder to train). My own shaky estimate based on a simple PyTorch profiling suggests that to achieve the same accuracy, FINN needs about 100,000 to 1,000,000 times more total FLOPs than FVM (if the stiffness matrix has been assembled, and we count the multigrid preconditioned CG FLOPs).
>
> Summary: I think this is a decent paper (perhaps not for ICLR), lots of experiments, good practice for a PhD student.

---

> > ### Public Comment · ~Matthias_Karlbauer1 · 2022-02-01
> > **Authors' response to Shuhao Cao's summary**
> >
> > Dear Shuhao Cao,
> >
> > We highly appreciate your comment on this paper and the clarification about its content.
> >
> > Certainly, an FVM solver can produce more accurate results in an ideal scenario (i.e. the toy problems that we have presented). Yet, in situations where the data is noisy, sparse, irregularly sampled over time, or crucial components such as a retardation function are unknown---that is in basically any real-world application---the necessary parametrizations and approximations make FVM inferior to our FINN approach. This was demonstrated on the real-world dataset, where FINN outperformed FVM.
> >
> > Moreover, and most importantly, there are no means of revealing unknown retardation functions, diffusion coefficients, boundary conditions, or other constitutive relationships when applying FVM. Instead, they always require tedious handcrafting, expert experience and knowledge, and lots of experimentation, whereas FINN determines and reveals those factors on a purely data-driven manner.
> >
> > No question, FVM models are beautiful and tremendously powerful when they are in the right hand and sufficient process knowledge is available. FINN is not a surrogate to solve PDEs faster or more efficiently, that is not our focus. We propose a model that can approximate partially observable dynamics with higher precision and can help modelers better understand the process at hand.
> >
> > Please let us know if you disagree in some argument.
> >
> > Best regards
> > The authors